# Genome-wide association and HLA region fine-mapping studies identify susceptibility loci for multiple common infections

Chao Tian[1], Bethann S. Hromatka[1], Amy K. Kiefer[1], Nicholas Eriksson[1], Suzanne M. Noble[2,3], Joyce Y. Tung[1] & David A. Hinds[1]

Infectious diseases have a profound impact on our health and many studies suggest that host genetics play a major role in the pathogenesis of most of them. We perform 23 genome-wide association studies for common infections and infection-associated procedures, including chickenpox, shingles, cold sores, mononucleosis, mumps, hepatitis B, plantar warts, positive tuberculosis test results, strep throat, scarlet fever, pneumonia, bacterial meningitis, yeast infections, urinary tract infections, tonsillectomy, childhood ear infections, myringotomy, measles, hepatitis A, rheumatic fever, common colds, rubella and chronic sinus infection, in over 200,000 individuals of European ancestry. We detect 59 genome-wide significant ($P < 5 \times 10^{-8}$) associations in genes with key roles in immunity and embryonic development. We apply fine-mapping analysis to dissect associations in the human leukocyte antigen region, which suggests important roles of specific amino acid polymorphisms in the antigen-binding clefts. Our findings provide an important step toward dissecting the host genetic architecture of response to common infections.

[1] 23andMe, Mountain View, CA 94041, USA. [2] Department of Microbiology & Immunology, UCSF, San Francisco, CA 94143, USA. [3] Department of Medicine, Division of Infectious Diseases, UCSF, San Francisco, CA 94143, USA. Correspondence and requests for materials should be addressed to C.T. (email: ctian@23andme.com) or to D.A.H. (email: dhinds@23andme.com)

Infectious diseases, the second leading cause of death world-wide, represent persistent challenges to human health due to increasing resistance to established treatments, lack of life-saving vaccines and medications in developing countries, and the ongoing emergence of new pathogens[1]. Studies have linked susceptibility to infectious agents to cancers, autoimmune diseases and drug hypersensitivity. Human papillomaviruses have been associated with multiple cancers[2]; rubella and mumps infection with type 1 diabetes (T1D) in children[3]; and Varicella zoster (shingles) with multiple sclerosis[4]. Reactivation of chronic persistent human herpesviruses has been linked to drug-induced hypersensitivity[5]. Thus, infectious diseases have a profound impact on our health, both directly as well as through connections with other diseases. Nevertheless, genetic studies of common infectious diseases have lagged behind those of other major complex diseases. In particular, only a few genome-wide association studies (GWASes) have been undertaken for common infectious diseases of lower mortality or for infectious diseases for which vaccines are available (Supplementary Table 1). Previous work has linked variants in human leukocyte antigen (HLA) region to shingles defined from electronic medical record data[6], to the risk for chronic hepatitis B (CHB) virus infection in Asian populations[7] and to the Epstein-Barr virus infection in Mexican American families[8]. Outside of HLA region, a variant near *HORMAD2* at 22q12.2 (rs2412971, $P = 1.48 \times 10^{-9}$) was recently reported to be associated with tonsillectomy[9]. Studies in a west African population have reported associations between two variants (rs4331426 and rs2057178) and susceptibility to tuberculosis (TB)[10, 11], and a later study found that the association with rs4331426 was not significant in an Asian population[12]. A prior study established associations between meningitis and variants in complement factor H (*CFH*; rs1065489) and in *CFH*-related protein 3 (*CFHR*; rs426736)[13]. *Neisseria meningitidis* is known to evade complement-mediated killing by the binding of host *CFH* to the meningococcal factor H-b-bind protein (fHbp).

We conducted 23 imputed GWASes for 23 common infections and infection-associated procedures (Table 1) with data drawn from more than 200,000 23andMe research participants of European ancestry who completed a standardized questionnaire about their history of infections (Supplementary Notes 1 and 2). In this study, we detected 59 genome-wide significant (GWS) associations and in some instances our findings replicated previous GWASes. We further imputed and tested the HLA classical alleles and amino acid polymorphisms to dissect independent HLA signals for multiple common infections. Our fine-mapping studies identify novel HLA associations in European population and suggest important roles of specific amino acid polymorphisms in the antigen-binding clefts. By studying multiple infections together, we find that virus-induced infectious diseases are mainly associated with class I molecules and the bacteria-induced infectious diseases are mainly associated with class II molecules, while with notable exceptions, as discussed below.

## Results

**Study summary.** In total, 59 GWS regions ($P < 5 \times 10^{-8}$) were discovered from the 23 imputed GWASes for 17 common infections we studied (Table 1). The diseases phenotypes were described in Supplementary Notes 1 and 2. Summary information for the index single-nucleotide polymorphism (SNP with lowest $P$ value) is presented in Tables 2 and 3, and GWAS results are displayed as Manhattan plots in Supplementary Fig. 1. All association results were adjusted for the corresponding genomic control inflation factors, and the Q–Q plots do not exhibit significant inflation in the $P$ values obtained (Supplementary Fig. 1). Regional association plots illustrate patterns of association for the index SNP and other variants within each associated genomic region (Supplementary Fig. 2). The strength of association in the region, previous reports of associations from the National Human Genome Research Institute GWAS Catalog entries, nearby expression quantitative trait loci (eQTL) and coding SNPs are discussed below and are also summarized in Supplementary Tables 3 and 5. The HLA region is significantly associated ($P < 5 \times 10^{-8}$) with 13 common infections, and we dissect independent signals in each association (Fig. 1 and

### Table 1 Discovery cohort characteristics

**Phenotypes with significant GWAS findings**

|  | Chickenpox | Shingles | Cold sores | Mononucleosis |
| --- | --- | --- | --- | --- |
| Cases | 107,769 | 16,711 | 25,108 | 17,457 |
| Control | 15,982 | 118,152 | 63,332 | 68,446 |
|  | Mumps | Hepatitis B | Plantar warts | Positive TB |
| Cases | 31,227 | 1425 | 24,994 | 4426 |
| Control | 54,153 | 218,180 | 37,451 | 84,290 |
|  | Strep throat[a] | Scarlet fever | Pneumonia | Bacterial meningitis |
| Cases | [qt] 52,487 | 6812 | 40,600 | 842 |
| Control | 22,017 | 113,837 | 90,039 | 82,778 |
|  | Yeast infections[a] | UTI frequency[a] | Tonsillectomy | Childhood ear infection |
| Cases | [qt] 52,218 | [qt] 35,000 | 60,098 | 46,936 |
| Control | 10,235 | 33,478 | 113,323 | 74,874 |
|  | Myringotomy |  |  |  |
| Cases | 4138 |  |  |  |
| Control | 85,089 |  |  |  |

**Phenotypes with no significant GWAS findings**

|  | Measles | Hepatitis A | Rheumatic fever | No. of colds last year[a] |
| --- | --- | --- | --- | --- |
| Cases | 38,219 | 2442 | 1115 | [qt] 43,826 |
| Control | 47,279 | 217,137 | 88,076 | 15,720 |
|  | Rubella | Chronic sinus infection |  |  |
| Cases | 12,000 | 5291 |  |  |
| Control | 71,597 | 79,622 |  |  |

*GWAS* genome-wide association study, *qt* quantitative traits, *UTI* urinary tract infection
[a]Strep throat, yeast infections, UTI frequency and number of colds last year are quantitative traits

**Table 2 Genome-wide significant associations for each disease part 1**

| Phenotype | Cytoband | Gene context | Variants | Alleles[a] | Freq | OR/effect[b] | (95% CI) | P value |
|---|---|---|---|---|---|---|---|---|
| Chickenpox | 6p21.33 | HLA | rs9266089 | G/A | 0.85 | 1.12 | (1.1–1.14) | 1.00E-10 |
| Shingles | 6p21.33 | HLA | rs2523591 | G/A | 0.58 | 1.14 | (1.13–1.16) | 1.74E-27 |
| | 9p21.3 | IFNA21 | rs7047299 | A/G | 0.56 | 1.07 | (1.06–1.09) | 1.67E-08 |
| Cold sores | 6p21.33 | HLA | rs885950 | C/A | 0.42 | 1.08 | (1.07–1.09) | 7.47E-13 |
| Mononucleosis | 6p21.33 | HLA | rs2596465 | T/C | 0.47 | 1.08 | (1.06–1.09) | 2.48E-09 |
| Mumps | 19q13.33 | FUT2 | rs516316 | C/G | 0.48 | 1.25 | (1.24–1.27) | 9.63E-72 |
| | 6p22.1 | HLA | rs114193679 | G/C | 0.99 | 1.72 | (1.62–1.84) | 2.23E-17 |
| | 14q32.2 | C14orf132--[]--BDKRB2 | rs11160318 | G/A | 0.68 | 1.1 | (1.08–1.11) | 4.56E-12 |
| | 11q24.2 | ST3GAL4 | rs3862630 | T/C | 0.12 | 1.13 | (1.1–1.15) | 1.21E-08 |
| Hepatitis A | 19q13.2 | IFNL4 | rs66531907 | A/C | 0.21 | 1.23 | (1.14–1.33) | 5.70E-08 |
| Hepatitis B | 6p21.32 | HLA | rs9268652 | G/A | 0.74 | 1.32 | (1.25–1.38) | 3.14E-09 |
| Plantar warts | 6p21.32 | HLA | rs9272050 | G/A | 0.38 | 1.19 | (1.17–1.21) | 9.66E-31 |
| | 1q21.3 | CRCT1--[]--LCE3E | rs6692209 | T/C | 0.62 | 1.08 | (1.06–1.09) | 5.25E-09 |
| Positive TB test | 6p21.32 | HLA | rs2894257 | C/G | 0.53 | 1.36 | (1.33–1.39) | 8.16E-36 |
| Strep throat | 6p21.33 | HLA | rs1055821 | T/G | 0.06 | 0.08 | (0.06–0.09) | 7.69E-11 |
| | 1p36.23 | ERRFI1---[]---SLC45A1 | rs35395352 | D/I | 0.72 | 0.03 | (0.03–0.04) | 3.90E-08 |
| Scarlet fever | 6p21.32 | HLA | rs36205178 | C/T | 0.81 | 1.24 | (1.21–1.28) | 9.49E-14 |
| Pneumonia | 6p21.33 | HLA | rs3131623 | T/A | 0.85 | 1.1 | (1.09–1.12) | 1.99E-15 |
| Bacterial meningitis | 17q21.33 | CA10 | rs1392935 | G/A | 0.91 | 1.78 | (1.6–1.99) | 1.19E-08 |
| Yeast infections | 18q12.1 | DSG1 | rs200520431 | D/I | 0.07 | 0.11 | (0.09–0.12) | 1.87E-16 |
| | 14q23.1 | PRKCH | rs2251260 | T/C | 0.17 | 0.05 | (0.04–0.06) | 3.46E-10 |
| | 14q32.2 | []---C14orf177 | rs7161578 | T/C | 0.39 | 0.04 | (0.03–0.04) | 4.04E-08 |
| UTI frequency | 8q24.3 | JRK-[]-PSCA | rs2976388 | G/A | 0.56 | 0.04 | (0.04–0.05) | 3.27E-10 |
| | 15q15.3 | FRMD5 | rs146906133 | T/C | 1.00 | 0.38 | (0.32–0.45) | 2.02E-08 |
| Childhood ear Infections | 19q13.33 | FUT2 | rs681343 | C/T | 0.52 | 1.11 | (1.1–1.12) | 3.51E-30 |
| | 22q11.21 | TBX1 | rs1978060 | G/A | 0.60 | 1.09 | (1.08–1.1) | 1.17E-19 |
| | 10p12.1 | RAB18--[]--MKX | rs2808290 | C/T | 0.51 | 1.07 | (1.07–1.08) | 5.09E-16 |
| | 15q26.2 | SPATA8---[]---LINC00923 | rs7174062 | G/A | 0.73 | 1.08 | (1.07–1.09) | 3.49E-14 |
| | 6p21.32 | HLA | rs4329147 | T/C | 0.83 | 1.11 | (1.09–1.13) | 9.55E-12 |
| | 9q34.2 | ABO | rs8176643 | D/I | 0.36 | 1.06 | (1.05–1.07) | 3.67E-11 |
| | 2p16.1 | EFEMP1 | rs1802575 | G/C | 0.87 | 1.09 | (1.07–1.1) | 1.47E-10 |
| | 2p24.1 | NT5C1B-RDH14---[]---OSR1 | rs5829676 | D/I | 0.40 | 1.06 | (1.05–1.07) | 1.78E-10 |
| | 11q13.3 | FGF3---[]--ANO1 | rs72931768 | G/C | 0.88 | 1.09 | (1.07–1.1) | 2.63E-09 |
| | 7q11.22 | AUTS2 | rs35213789 | C/T | 0.74 | 1.06 | (1.05–1.07) | 3.75E-09 |
| | 7q22.3 | CDHR3 | rs114947103 | C/T | 0.18 | 1.07 | (1.06–1.08) | 5.40E-09 |
| | 8q22.2 | NIPAL2--[]--KCNS2 | rs13281988 | C/G | 0.31 | 1.06 | (1.05–1.07) | 9.84E-09 |
| | 3p21.31 | BSN | rs67035515 | I/D | 0.18 | 1.07 | (1.05–1.08) | 1.56E-08 |
| | 6q26 | PLG | rs73015965 | G/A | 0.01 | 1.43 | (1.34–1.53) | 3.78E-08 |
| Myringotomy | 22q11.21 | TBX1 | rs1978060 | G/A | 0.60 | 1.17 | (1.14–1.2) | 3.34E-10 |

HLA human leukocyte antigen, SNP single-nucleotide polymorphism, TB tuberculosis, UTI urinary tract infection
[a]The alleles for each SNP are represented as high-risk allele/low-risk allele
[b]Strep throat, plantar warts and UTI frequency are quantitative traits, for which we reported the effect size instead of the odds ratio

Tables 4 and 5, see also Supplementary Fig. 3 for HLA regional plots).

We estimated the genetic heritability and correlations of these common infections using the GWAS data. The proportion of variance attributable to genome-wide SNPs (SNP heritability) was estimated to be 6% on average for the common infections. Multiple infections showed significant positive genetic correlations ($P < 2 \times 10^{-4}$, Supplementary Fig. 4). The results are presented in Supplementary Tables 6 and 7 and discussed in detail in Supplementary Notes 3 and 4.

**Chickenpox and shingles**. We identified independent associations of adults with a history of chickenpox with HLA-A and HLA-B in the class I region. On conditional analysis within HLA-A, the amino acid polymorphism at position 107 of HLA-A protein (HLA-A Gly107, $P = 3.77 \times 10^{-10}$, odds ratio (OR) = 1.09) accounted for the HLA allele association (HLA-A*02:01, $P = 1.08 \times 10^{-9}$, OR = 0.92, conditional $P$ value is 0.90) in this interval. Although HLA-A Gly107 is not located in the peptide-binding cleft, it is in high linkage disequilibrium (LD) with HLA-A Asp74, HLA-A Gly62 and HLA-A V95 ($r^2 > 0.95$), which are all

in the peptide-binding cleft. In HLA-B, only rs9266089 ($P = 1.00 \times 10^{-10}$, OR = 1.12) met the threshold for GWS.

We identified multiple independent HLA signals for shingles. Within HLA-A, the amino acid polymorphism HLA-A Arg97 ($P = 2.75 \times 10^{-22}$, OR = 0.88), located in the peptide-binding groove, accounted for most of the SNP effect (rs2523815, $P = 1.83 \times 10^{-22}$, OR = 1.12, conditional $P$ value is 0.04) and the HLA allele effect (HLA-A*02:01, $P = 8.91 \times 10^{-20}$, OR = 0.88, conditional $P$ value is 0.06). HLA-A*02:01 is implicated in both our chickenpox and shingles GWASes. The amino acid associations are different, but HLA-A Arg97 is in high LD with HLA-A Gly107 ($r^2 = 0.73$). Within HLA-B, rs2523591 ($P = 1.74 \times 10^{-27}$, OR = 1.14) had a stronger association than HLA classical variants. There was a significant residual effect for rs2523591 after conditioning on HLA allele (conditional $P$ value is $1.71 \times 10^{-9}$) or amino acid (conditional $P$ value is $3.13 \times 10^{-5}$) associations in HLA-B region. The SNP rs2523591 is in LD with rs9266089 ($r^2 = 0.24$, D' = 0.98), which is a GWS signal in chickenpox GWAS. After adjusting for the signals in HLA-A and HLA-B regions, we observed independent secondary signals in the class III region (rs41316748, $P = 1.44 \times 10^{-12}$, OR = 1.19) and in the class II region index by HLA-DRB1 PheSerHis13

**Table 3 Genome-wide significant associations for each disease part 2**

| Phenotype | Cytoband | Gene context | Variants | Alleles[a] | Freq | OR/effect[b] | (95% CI) | P value |
|---|---|---|---|---|---|---|---|---|
| Tonsillectomy | 12p13.31 | LTBR | rs10849448 | A/G | 0.25 | 1.13 | (1.12–1.14) | 2.35E-35 |
| | 22q12.2 | MTMR3 | rs201112509 | I/D | 0.69 | 1.12 | (1.11–1.13) | 3.39E-34 |
| | 17p11.2 | TNFRSF13B | rs34557412 | G/A | 0.01 | 1.59 | (1.52–1.67) | 2.66E-21 |
| | 6p21.33 | HLA | rs41543314 | G/A | 0.03 | 1.23 | (1.2–1.26) | 5.36E-21 |
| | 13q33.3 | TNFSF13B | rs200748895 | D/I | 0.03 | 1.26 | (1.23–1.29) | 1.20E-17 |
| | 7p12.3 | IGFBP3---[] | rs80077929 | T/C | 0.11 | 1.11 | (1.1–1.13) | 1.74E-15 |
| | 4q24 | NFKB1 | rs230523 | T/C | 0.67 | 1.07 | (1.06–1.08) | 4.54E-14 |
| | 7p15.2 | HOXA1-[]-HOXA2 | rs6668 | T/C | 0.37 | 1.06 | (1.06–1.07) | 1.71E-13 |
| | 14q21.1 | MIPOL1-[]--FOXA1 | rs148131694 | T/C | 0.44 | 1.07 | (1.06–1.08) | 7.86E-13 |
| | 4q24 | TET2 | rs1391439 | G/A | 0.40 | 1.06 | (1.05–1.07) | 2.76E-12 |
| | 2p14 | SPRED2 | rs201473667 | D/I | 0.47 | 1.06 | (1.05–1.07) | 5.39E-12 |
| | 20q13.12 | NCOA5--[]-CD40 | rs6032664 | A/T | 0.26 | 1.07 | (1.06–1.08) | 7.31E-12 |
| | 7p12.2 | C7orf72---[]--IKZF1 | rs876037 | T/A | 0.68 | 1.06 | (1.05–1.07) | 1.12E-11 |
| | 13q21.33 | KLHL1 | rs9542155 | T/C | 0.32 | 1.06 | (1.05–1.07) | 1.92E-11 |
| | 7p22.2 | GNA12 | rs2644312 | G/A | 0.71 | 1.06 | (1.05–1.07) | 2.73E-11 |
| | 3q21.2 | SLC12A8 | rs1980080 | C/T | 0.33 | 1.06 | (1.05–1.07) | 3.16E-11 |
| | 16p11.2 | SBK1 | rs141876325 | I/D | 0.28 | 1.06 | (1.05–1.07) | 4.72E-11 |
| | 1q41 | DUSP10---[]---HHIPL2 | rs12126292 | G/T | 0.12 | 1.09 | (1.07–1.1) | 7.21E-11 |
| | 4p15.2 | LGI2--[]-SEPSECS | rs10939037 | A/G | 0.56 | 1.05 | (1.05–1.06) | 1.42E-10 |
| | 12q24.12 | SH2B3 | rs3184504 | T/C | 0.50 | 1.05 | (1.04–1.06) | 3.07E-10 |
| | 4q21.1 | CXCL13 | rs7685785 | C/T | 0.89 | 1.09 | (1.07–1.1) | 4.52E-10 |
| | 7q31.2 | MDFIC---[]---TFEC | rs2023703 | C/A | 0.33 | 1.05 | (1.05–1.06) | 7.79E-10 |
| | 19p13.2 | ADAMTS10---[]--ACTL9 | rs2918308 | A/C | 0.84 | 1.07 | (1.06–1.08) | 1.10E-09 |
| | 17p13.3 | RAP1GAP2 | rs67968065 | D/I | 0.52 | 1.05 | (1.04–1.06) | 2.08E-09 |
| | 21q22.11 | ITSN1 | rs200746495 | D/I | 0.87 | 1.07 | (1.06–1.09) | 4.45E-09 |
| | 2q33.3 | ADAM23 | rs1448903 | G/A | 0.09 | 1.08 | (1.07–1.1) | 5.13E-09 |
| | 1p36.23 | RERE--[]--ENO1 | rs12068123 | G/A | 0.45 | 1.05 | (1.04–1.06) | 5.60E-09 |
| | 11p15.4 | ST5 | rs11042055 | A/G | 0.43 | 1.05 | (1.04–1.06) | 6.96E-09 |
| | 20q13.33 | SAMD10 | rs41278232 | G/A | 0.89 | 1.08 | (1.06–1.09) | 7.54E-09 |
| | 9q34.2 | ABO-[]--SURF6 | rs635634 | T/C | 0.20 | 1.06 | (1.05–1.07) | 8.47E-09 |
| | 17q11.2 | FOXN1 | rs62066768 | A/G | 0.08 | 1.09 | (1.07–1.11) | 1.09E-08 |
| | 6q23.3 | OLIG3---[]---TNFAIP3 | rs11757201 | C/G | 0.20 | 1.06 | (1.05–1.07) | 2.97E-08 |
| | 16p11.2 | MAPK3--[]--CORO1A | rs12931792 | G/A | 0.54 | 1.06 | (1.05–1.07) | 4.06E-08 |
| | 7q31.2 | WNT2 | rs200608253 | I/D | 0.52 | 1.05 | (1.04–1.05) | 4.07E-08 |
| | 22q11.21 | TBX1 | rs41298830 | A/C | 0.77 | 1.06 | (1.05–1.07) | 4.95E-08 |

CI confidence interval, HLA human leukocyte antigen, OR odds ratio, SNP single-nucleotide polymorphism, UTI urinary tract infection
[a]The alleles for each SNP are represented as high-risk allele/low-risk allele
[b]Strep throat, plantar warts and UTI frequency are quantitative traits, for which we reported the effect size instead of the OR

($P = 5.9 \times 10^{-10}$, OR = 0.92), which is in the peptide-binding cleft of HLA-DRB1.

We identified a variant upstream of IFNA21 (rs7047299, $P = 1.67 \times 10^{-8}$, OR = 1.07) as a GWS association with shingles. None of the variants within 500 kb and in moderate LD ($r^2 > 0.6$) with rs7047299 were coding, nor were they reported as eQTL. IFNA21 encodes type I interferon and is mainly involved in innate immune response against viral infection. It has been shown to be involved in the pathogenesis of rubella[14] and may also influence susceptibility to asthma and atopy[15].

**Cold sores.** We identified independent associations between two SNPs in the class I region and a history of cold sores. One is 2 kb upstream of POU5F1 indexed by rs885950 ($P = 7.47 \times 10^{-13}$, OR = 1.08) and the other is upstream of HCP5 indexed by rs4360170 ($P = 3.41 \times 10^{-9}$, OR = 1.13). We also found a GWS association with HLA-B-ThrGly45 in the peptide-binding cleft of the HLA-B protein ($P = 4.91 \times 10^{-12}$, OR = 0.91). Upon conditioning on HLA-B ThrGly45, the two SNPs had significant residual effects (conditional $P$ values are $6.28 \times 10^{-9}$ and $1.32 \times 10^{-5}$). However, the effect of HLA-B ThrGly45 was largely removed (conditional $P$ value is 0.001) after conditioning on the two SNPs.

**Mononucleosis.** We identified a variant upstream of HCP5 in the HLA class I region as a GWS association (indexed by rs2596465, $P = 2.48 \times 10^{-9}$, OR = 1.08). No GWS HLA classical allele or amino acid variant was identified.

**Mumps.** We identified four independent GWS associations with mumps. Variation in HLA-A, in the class I region, was significantly associated with mumps. The amino acid polymorphism HLA-A Gln43 ($P = 2.51 \times 10^{-17}$, OR = 1.71) accounted for most of the SNP association (rs114193679, $P = 2.23 \times 10^{-17}$, OR = 1.72, conditional $P$ value is 0.19) and the HLA allele association (HLA-A*02:05, $P = 1.73 \times 10^{-17}$, OR = 0.56, conditional $P$ value is 0.26) in the HLA-A region.

FUT2 (rs516316, $P = 9.63 \times 10^{-72}$, OR = 1.25) and ST3GAL4 (rs3862630, $P = 1.21 \times 10^{-8}$, OR = 1.13) also have highly significant associations with mumps. Both are components of the glycosphingolipid (GSL) biosynthesis pathway, which was the most significant Meta-Analysis Gene-set Enrichment of variaNT Associations (MAGENTA) -analyzed pathway (Table 6: $P = 1 \times 10^{-4}$ and false discovery rate (FDR) = 0.003) identified using mumps GWAS data. This pathway also includes the ABO gene (rs8176643, $P = 5.81 \times 10^{-5}$, OR = 1.07). GSLs are found in cell membranes of organisms ranging from bacteria to humans and play important biological roles in membrane structure, host–pathogen interactions, cell–cell recognition and modulation of membrane protein function. Genetic variation in human FUT2 determines whether ABO blood group antigens are secreted into body fluids[16]. Detailed functional annotations of the genes

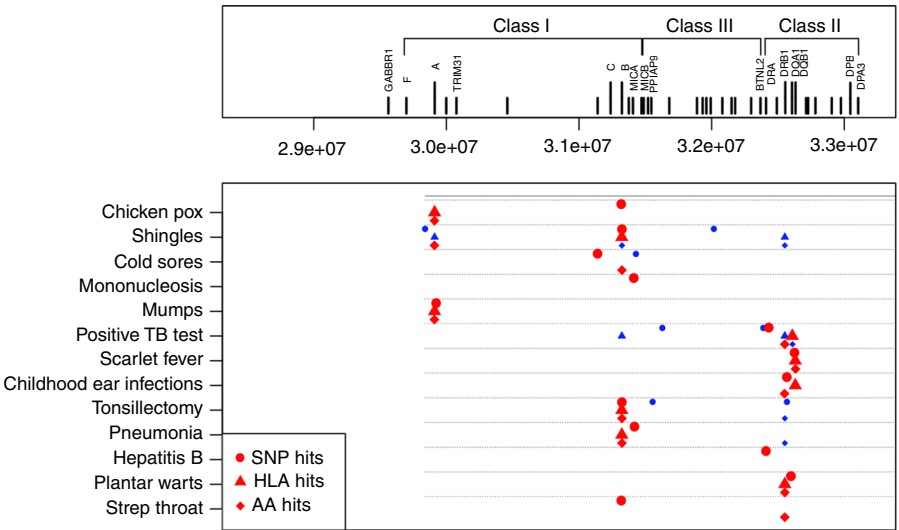

**Fig. 1** Summary of independent HLA signals. The strongest associated signals (*red*) and putative independent secondary signals (*blue*) are shown for each disease along with their location within the region. Independent secondary signals were defined as those with residual conditional association using the same significant threshold as the primary association signal mentioned in the main text

identified in our mumps GWAS are provided in Supplementary table 3. The high-risk allele rs516316 (C) is in complete LD with rs601338 (A), a nonsense variant in *FUT2* that encodes the 'non-secretor' (se) allele, which has been reported to provide resistance to Norovirus[17] and upper respiratory infections[18] ('non-secretor' allele is linked to lower rate of childhood ear infections in our GWAS, Supplementary Fig. 5A), and increase risk to Crohn's disease[74] and T1D[19]. These results are interesting in light of studies showing that the number of T1D cases increase significantly after a mumps epidemic, suggesting that mumps virus might promote autoimmunity and trigger T1D[3]. The surface glycoprotein of mumps virus, hemagglutinin-neuraminidase, attaches to sialic acid receptors and promotes fusion and viral entry into host cells[20]. ABO antigens can modulate the interaction between pathogens and cell surface sialic acid receptors[21]; this presents a plausible mechanism by which secretor status could modify susceptibility to mumps infections. *ST3GAL4* encodes a member of the glycosyltransferase 29 family, a group of enzymes involved in protein glycosylation.

We also identified a significant association between mumps and a variant on chromosome 14 (rs11160318, $P = 4.56 \times 10^{-12}$, OR = 1.1) that is located in the intergenic region upstream of *BDKRB2*. The role of *BDKRB2* in mumps susceptibility is not clear, but the *BDKRB2* product has high affinity for intact kinins, which mediate a wide spectrum of biological effects including inflammation, pain, vasodilation, and smooth muscle contraction and relaxation[22].

**Hepatitis A**. Our GWAS showed a suggestive association with *IFNL4* (rs66531907, $P = 5.7 \times 10^{-8}$, OR = 1.23, 1 kb upstream of *IFNL4*). *IFNL4* is located upstream of *IFNL3* (also known as *IL28B*). The high-risk allele rs66531907 (C) in our hepatitis A GWAS is in almost complete LD with rs8099917 (T) ($r^2 = 0.992$), which has been associated with progression to chronic hepatitis C infection (HCV) and poor response to HCV therapy in multiple studies[23].

**Hepatitis B**. Our GWAS of hepatitis B virus (HBV) infection in Europeans identified a GWS association with rs9268652 ($P = 3.14 \times 10^{-9}$, OR = 1.32, in *HLA-DRA*), consistent with the association of the HLA class II region with CHB susceptibility. Fine-mapping analysis failed to identify GWS associations with

HLA classical variants, but *DQB1*06:02* was moderately associated with hepatitis B ($P = 5.47 \times 10^{-6}$, OR = 0.75).

**Plantar warts**. We identified two GWS associations with plantar wart, the first of which was with the *HLA-DRB1* in the class II region. The amino acid polymorphism *HLA-DRB1 PheSerTyr13* ($P = 1.84 \times 10$-39, OR = 0.85), in the peptide-binding clefts of *HLA-DRB1*, was highly significant. Conditioning on it, the effects of *HLA-DRB1*15:01* ($P = 4.48e$-14, OR = 1.14, conditional $P$ value is 0.36), *HLA- DRB1*04:01* ($P = 4.88 \times 10^{-18}$, OR = 1.21, conditional $P$ value is 0.001), and rs9272050 ($P = 9.66 \times 10^{-31}$, OR = 1.19, conditional $P$ value is 0.06) were largely removed, but a small residual effect remained for rs28752534 ($P = 1.50 \times 10^{-30}$, OR = 1.11, conditional $P$ value is $2.15 \times 10^{-8}$).

We also identified a GWS association with rs6692209 ($P = 5.25 \times 10^{-9}$, OR = 1.08) near *LCE3E* in the Epidermal Differentiation Complexn 1q21. The EDC comprises a remarkable density of gene families that determine differentiation of the human epidermis[24]. The high-risk allele rs6692209 (T) is in high LD ($r^2 = 0.86$) with rs2105117 (A) ($P = 2.98 \times 10^{-8}$, OR = 1.07, in our GWAS), a missense variant in *LCE5A*. Variants in late cornified envelope (LCE) genes have been implicated in GWASes of atopic dermatitis (rs3126085-A, $P = 6 \times 10^{-12}$, $r^2 = 0.047$ with rs6692209)[25] and psoriasis (rs4085613-A, $P = 7 \times 10^{-30}$, $r^2 = 0.002$ with rs6692209)[26], although the index SNPs identified in those GWASes were in low LD with our identified region. While studies have linked the EDC to various skin diseases, the role of the LCE proteins in epidermal biology has not been extensively studied. One study suggested that LCE proteins of groups 1, 2, 5 and 6 are involved in normal skin barrier function, while *LCE3* genes encode proteins that are involved in barrier repair after injury or inflammation[27].

**Positive tuberculosis test**. We identified multiple independent associations between a history of positive TB skin test and HLA. In the most significant *HLA-DRA-DQA1* region, conditioning on *HLA-DRB1 Leu67* ($P = 1.14 \times 10^{-29}$, OR = 1.3) and *HLA-DQA1 His129* ($P = 1.56 \times 10^{-13}$, OR = 1.19) removed the effect of *DQA1*01:02* ($P = 7.04 \times 10^{-22}$, OR = 0.75, conditional $P$ value is 0.54), *DQA1*03:01* ($P = 3.99 \times 10^{-10}$, OR = 1.27, conditional $P$ value is 0.36), and rs3135359 ($P = 8.82 \times 10^{-21}$, OR = 1.21, conditional $P$ value is 0.06); however, the effect of rs2894257

**Table 4 HLA fine-mapping results for each disease part 1**

| Phenotype | Gene | Cytoband | Variants[a] | Alleles[b] | Freq | OR/effect[c] | (95% CI) | Con. P value | P value |
|---|---|---|---|---|---|---|---|---|---|
| Chickenpox | HLA-A | 6p22.1 | HLA-A Gly107 | GW | 0.71 | 1.09 | (1.07–1.1) | 3.77E-10 | 3.77E-10 |
| | | 6p22.1 | HLA-A*02:01 | NA | 0.27 | 0.92 | (0.91–0.93) | 1.08E-09 | 1.08E-09 |
| | HLA-B | 6p21.33 | rs9266089 | G/A | 0.85 | 1.12 | (1.1–1.14) | 1.00E-10 | 1.00E-10 |
| Shingles | HLA-A | 6p22.1 | rs2523815 | A/G | 0.64 | 1.12 | (1.11–1.14) | 6.17E-20 | 1.83E-22 |
| | | 6p22.1 | HLA-A Arg97 | IRM | 0.36 | 0.88 | (0.87–0.9) | 2.75E-22 | 2.75E-22 |
| | | 6p22.1 | HLA-A*02:01 | NA | 0.27 | 0.88 | (0.87–0.89) | 1.72E-12 | 8.91E-20 |
| | HLA-B | 6p21.33 | HLA-B*44:02 | NA | 0.09 | 0.81 | (0.79–0.82) | 3.54E-23 | 3.54E-23 |
| | | 6p21.33 | HLA-B*57:01 | NA | 0.04 | 1.22 | (1.18–1.25) | 1.03E-08 | 1.81E-10 |
| | | 6p21.33 | HLA-B Trp147 | WL | 0.95 | 0.92 | (0.89–0.94) | 6.67E-10 | 1.24E-03 |
| | | 6p21.33 | HLA-B GluMet45 | EMTKG | 0.52 | 1.11 | (1.1–1.13) | 9.51E-16 | 1.65E-18 |
| | | 6p21.33 | rs2523591 | G/A | 0.58 | 1.14 | (1.13–1.16) | 1.74E-27 | 1.74E-27 |
| | TNXB | 6p21.33 | rs41316748 | C/T | 0.04 | 1.19 | (1.15–1.23) | 1.02E-08 | 1.44E-12 |
| | | 6p21.32 | HLA-DRB1*11:01 | NA | 0.06 | 0.79 | (0.77–0.82) | 9.84E-10 | 2.96E-11 |
| | | 6p21.32 | HLA-DRB1 PheSerHis13 | FSHYGR | 0.66 | 0.92 | (0.91–0.94) | 1.22E-09 | 5.90E-10 |
| Cold sores | POU5F1-[]--HCG27 | 6p21.33 | rs885950 | C/A | 0.42 | 1.08 | (1.07–1.09) | 7.47E-13 | 7.47E-13 |
| | HLA-B | 6p21.33 | HLA-B ThrGly45 | EMTKG | 0.23 | 0.91 | (0.9–0.92) | 4.91E-12 | 4.91E-12 |
| | MICA--[]--MICB | 6p21.33 | rs4360170 | A/G | 0.91 | 1.13 | (1.11–1.16) | 3.38E-09 | 3.41E-12 |
| Mononucleosis | MICA--[]--MICB | 6p21.33 | rs2596465 | T/C | 0.47 | 1.08 | (1.06–1.09) | 2.48E-09 | 2.48E-09 |
| Mumps | HLA-A | 6p22.1 | HLA-A Gln43 | QR | 0.99 | 1.71 | (1.6–1.82) | 2.51E-17 | 2.51E-17 |
| | | 6p22.1 | HLA-A*02:05 | NA | 0.01 | 0.56 | (0.52–0.6) | 1.73E-17 | 1.73E-17 |
| | | 6p22.1 | rs114193679 | G/C | 0.99 | 1.72 | (1.62–1.84) | 2.23E-17 | 2.23E-17 |
| Hepatitis B | HLA-DRA | 6p21.32 | rs9268652 | G/A | 0.74 | 1.32 | (1.25–1.38) | 3.14E-09 | 3.14E-09 |
| Plantar warts | HLA-DRB1 | 6p21.32 | HLA-DRB1*04:01 | NA | 0.08 | 1.21 | (1.19–1.24) | 4.88E-18 | 4.88E-18 |
| | | 6p21.32 | HLA-DRB1*15:01 | NA | 0.13 | 1.14 | (1.12–1.15) | 3.50E-19 | 4.48E-14 |
| | | 6p21.32 | HLA-DRB1 PheSerTyr13 | FSHYGR | 0.63 | 0.85 | (0.84–0.86) | 1.84E-39 | 1.84E-39 |
| | | 6p21.32 | rs28752534 | T/C | 0.64 | 1.11 | (1.1–1.13) | 9.28E-09 | 1.50E-30 |
| | | 6p21.32 | rs9272050 | G/A | 0.38 | 1.19 | (1.17–1.21) | 9.65E-31 | 9.66E-31 |

CI confidence interval, HLA human leukocyte antigen, NA not applicable, OR odds ratio, SNP single-nucleotide polymorphism, UTI urinary tract infection
[a]We identify independent associations for each category of variants: SNP, HLA (classical allele) and AA (HLA amino acid). For amino acid polymorphism, the label specifies the amino acid and the position. For example, 'HLA-A Val95' means amino acid Val at position 95 of the HLA-A protein
[b]The alleles for each SNP are represented as risk allele/non-risk allele; the allele for the HLA classical allele is recorded as 'NA', which means it either has or does not have the tested allele; the alleles for HLA amino acid polymorphisms are all the possible amino acids (one-letter code) at that position
[c]Strep throat, plantar warts and UTI frequency are quantitative traits, for which we reported the effect size instead of the odds ratio
[d]The 'Cond. P value' column contains conditional P values generated after iterative conditional regression within each category of variants (SNP, HLA alleles and HLA amino acid polymorphisms). The 'P value' column contains the unconditional association test P values

($P = 8.16 \times 10^{-36}$, OR = 1.36, conditional $P$ value is $1.2 \times 10^{-7}$) remained moderately significant. Upon conditioning on the signals in the HLA-DRA-DQA1 region, secondary signals were detected in the class I (HLA-B*08:01, $P = 8.43 \times 10^{-14}$, OR = 1.29) and class III regions (rs148844907 in C6orf47, $P = 2.21 \times 10^{-12}$, OR = 1.92).

**Strep throat and scarlet fever**. We found a variant in HLA-B class I region (index SNP rs1055821, $P = 7.69 \times 10^{-11}$, effect = 0.08) as a GWS association with strep throat. HLA-B*57:01 was almost GWS ($P = 5.06 \times 10^{-8}$, effect = 0.08). HLA-DRB1_CysTyr30 in class II region also showed a significant effect ($P = 2.14 \times 10^{-9}$, effect = −0,04). Conditioning on rs1055821, the effect of HLA-B*57:01 was removed (conditional $P$ value is 0.84) and the effect of HLA-DRB1 CysTyr30 was substantially decreased (conditional $P$ value is 0.005).

We also found a GWS association with a small insertion, rs35395352, in 1p36.23 ($P = 3.90 \times 10^{-8}$, effect = 0.03). None of the variants within 500 kb and in moderate LD ($r^2 > 0.6$) with this SNP were coding. However, two variants have been reported as cis-eQTLs: rs7548511 ($P = 7.84 \times 10^{-8}$, effect = 0.03, risk allele A, $r^2 = 0.96$ with rs35395352) is an eQTL for ENO1 (eQTL $P$ value is $8 \times 10^{-5}$) in lymphoblastoid[28] and rs11121129 ($P = 9.62 \times 10^{-8}$, effect = 0.03, risk allele G, $r^2 = 0.96$ with rs35395352) is an eQTL for TNFRSF9 (eQTL $P$ value is $4.4 \times 10^{-4}$) in lymphoblastoid[29]. The product of ENO1, alpha enolase, has been identified as an autoantigen in Hashimoto's encephalopathy[30] and a putative autoantigen in severe asthma[31] and Behcet's disease[32]. The TNFRSF9-encoded protein is a member of the tumor necrosis factor (TNF) -receptor superfamily, which contributes to the clonal expansion, survival and development of T cells. The same 1p36.23 region has been implicated in GWASes of psoriasis (rs11121129, risk allele A, $P = 1.7 \times 10^{-8}$, $r^2 = 0.96$ with rs35395352)[33]. Interestingly, the strep throat risk allele rs11121129-A is protective against psoriasis. Studies found that exacerbation of chronic psoriasis can be associated with streptococcal throat infections, and the T cells generated by palatine tonsils can recognize skin keratin determinants in patients' blood[34].

We found a GWS association between scarlet fever and the HLA-DQB1 in class II region. The amino acid polymorphism HLA-DQB1 Gly45 ($P = 6.35 \times 10^{-14}$, OR = 1.19), located in the peptide-binding cleft of HLA-DQB1, accounted for the SNP association (rs36205178, $P = 9.49 \times 10^{-14}$, OR = 1.24, conditional $P$ values is 0.83) and the HLA allele association (HLA-DQB1*03:01, $P = 2.98 \times 10^{-14}$, OR = 0.83, conditional $P$ values is 0.14) in this region. Interestingly, we did not observe any cross disease effect between 'scarlet fever' and 'strep throat' in our analysis.

**Pneumonia**. We found a GWAS association between pneumonia and a variant in the HLA class I region (rs3131623, $P = 1.99 \times 10^{-15}$, OR = 1.1). Conditioning on rs3131623, the effects of HLA-B SerTrpAsn97 ($P = 2.37 \times 10^{-12}$, OR = 0.94, conditional $P$ value is $9.33 \times 10^{-4}$), in the peptide-binding cleft, was significantly reduced. HLA-DRB1 LS11 in class II region also showed a significant independent effect ($P = 4.10 \times 10^{-11}$, OR = 0.94). Conditioning on rs3131623, HLA-DRB1 LS11 remained moderately significant (conditional $P$ value is $3.89 \times 10^{-6}$).

**Table 5 HLA fine-mapping results for each disease part 2**

| Phenotype | Gene | Cytoband | Variants[a] | Alleles[b] | Freq | OR/ Effect[c] | (95% CI) | Con. P value | P value |
|---|---|---|---|---|---|---|---|---|---|
| Positive TB test | HLA-B | 6p21.33 | HLA-B*08:01 | NA | 0.11 | 1.29 | (1.24-1.33) | 1.73E-11 | 8.43E-14 |
| | C6orf47 | 6p21.33 | rs148844907 | A/T | 0.01 | 1.92 | (1.74-2.12) | 4.95E-10 | 2.21E-12 |
| | HLA-DRA- DQA1 | 6p21.32 | rs3135359 | C/T | 0.71 | 1.21 | (1.18-1.24) | 2.63E-13 | 8.82E-21 |
| | | 6p21.32 | rs2894257 | C/G | 0.53 | 1.36 | (1.33-1.39) | 8.16E-36 | 8.16E-36 |
| | | 6p21.32 | HLA-DRB1 Leu67 | LIF | 0.41 | 1.3 | (1.27-1.33) | 1.14E-29 | 1.14E-29 |
| | | 6p21.32 | HLA-DQA1*01:02 | NA | 0.20 | 0.75 | (0.73-0.78) | 7.04E-22 | 7.04E-22 |
| | | 6p21.32 | HLA-DQA1*03:01 | NA | 0.10 | 1.27 | (1.22-1.32) | 1.59E-09 | 3.99E-10 |
| | | 6p21.32 | HLA-DQA1 His129 | QHx | 0.66 | 1.19 | (1.16-1.22) | 2.40E-13 | 1.56E-13 |
| Strep throat | HLA-B | 6p21.33 | rs1055821 | T/G | 0.06 | 0.08 | (0.06-0.09) | 7.69E-11 | 7.69E-11 |
| | HLA-DRB1 | 6p21.32 | HLA-DRB1 CysTyr30 | CYLGRH | 0.83 | −0.04 | (−0.05-0.04) | 2.14E-09 | 2.14E-09 |
| Scarlet fever | HLA-DQB1 | 6p21.32 | rs36205178 | C/T | 0.81 | 1.24 | (1.21-1.28) | 9.49E-14 | 9.49E-14 |
| | | 6p21.32 | HLA-DQB1*03:01 | NA | 0.20 | 0.83 | (0.81-0.86) | 2.98E-14 | 2.98E-14 |
| | | 6p21.32 | HLA-DQB1 Gly45 | GE | 0.80 | 1.19 | (1.16-1.22) | 6.35E-14 | 6.35E-14 |
| Pneumonia | HLA-B | 6p21.33 | HLA-B*08:01 | NA | 0.11 | 0.91 | (0.9-0.92) | 2.87E-11 | 2.87E-11 |
| | | 6p21.33 | HLA-B SerTrpAsn97 | STWRNV | 0.34 | 0.94 | (0.93-0.95) | 2.37E-12 | 2.37E-12 |
| | MICA-- []--MICB | 6p21.33 | rs3131623 | T/A | 0.85 | 1.1 | (1.09-1.12) | 1.99E-15 | 1.99E-15 |
| | HLA-DRB1 | 6p21.32 | HLA-DRB1 LeuSer11 | LSVGDP | 0.52 | 0.94 | (0.94-0.95) | 1.42E-09 | 4.10E-11 |
| Tonsillectomy | HLA-B | 6p21.33 | rs41543314 | G/A | 0.03 | 1.23 | (1.2-1.26) | 5.36E-21 | 5.36E-21 |
| | | 6p21.33 | HLA-B*57:01 | NA | 0.04 | 1.22 | (1.19-1.24) | 6.29E-22 | 6.29E-22 |
| | | 6p21.33 | HLA-B Val97 | STWRNV | 0.04 | 1.21 | (1.19-1.24) | 1.44E-21 | 1.44E-21 |
| | LST1,LST1 | 6p21.33 | rs1052248 | A/T | 0.28 | 1.05 | (1.04-1.06) | 3.69E-08 | 3.15E-17 |
| | HLA-DRB1 | 6p21.32 | HLA-DRB1 LeuValGly11 | LSVGDP | 0.42 | 1.06 | (1.05-1.07) | 1.67E-09 | 1.02E-12 |
| | | 6p21.32 | rs140177540 | D/I | 0.47 | 1.06 | (1.05-1.07) | 2.92E-12 | 5.02E-15 |
| | GRM4-- []-HMGA1 | 6p21.31 | rs201220830 | D/I | 0.05 | 1.11 | (1.09-1.14) | 1.32E-08 | 3.65E-08 |
| Childhood ear infections | HLA-DRB1 | 6p21.32 | HLA-DRB1 Gln96 | EHYxQ | 0.17 | 0.92 | (0.91-0.93) | 2.11E-12 | 2.11E-12 |
| | | 6p21.32 | rs4329147 | T/C | 0.83 | 1.11 | (1.09-1.13) | 9.55E-12 | 9.55E-12 |
| | | 6p21.32 | HLA-DQB1*06:02 | NA | 0.13 | 0.92 | (0.91-0.94) | 5.79E-10 | 5.79E-10 |

CI confidence interval, HLA human leukocyte antigen, NA not applicable, OR odds ratio, SNP single-nucleotide polymorphism, UTI urinary tract infection

[a]We identify independent associations for each category of variants: SNP, HLA (classical allele) and AA (HLA amino acid). For amino acid polymorphism, the label specifies the amino acid and the position. For example, 'HLA-A Val95' means amino acid Val at position 95 of the HLA-A protein

[b]The alleles for each SNP are represented as risk allele/non-risk allele; the allele for the HLA classical allele is recorded as 'NA', which means it either has or does not have the tested allele; the alleles for HLA amino acid polymorphisms are all the possible amino acids (one-letter code) at that position

[c]Strep throat, plantar warts and UTI frequency are quantitative traits, for which we reported the effect size instead of the odds ratio

[d]The 'Cond. P value' column contains conditional P values generated after iterative conditional regression within each category of variants (SNP, HLA alleles and HLA amino acid polymorphisms). The 'P value' column contains the unconditional association test P values

**Bacterial meningitis**. We identified a GWS association with CA10 (carbonic anhydrase X) (rs1392935, $P = 1.19 \times 10^{-8}$, OR = 1.78, intronic of CA10). None of the variants within 500 kb and in moderate LD ($r^2 > 0.6$) with rs1392935 were coding, nor were they reported as an eQTL. However, rs1392935 falls within an enhancer that is defined in the fetal brain. The protein encoded by CA10 is an acatalytic member of the alpha-carbonic anhydrase subgroup and it is thought to play a role in the central nervous system, especially in brain development[35].

**Vaginal yeast infection**. We found GWS associations between vulvovaginal candidiasis and variants in DSG1 (rs200520431, $P = 1.87 \times 10^{-16}$, effect = 0.11). The index SNP rs200520431 is intronic and in high LD ($r^2 > 0.8$) with multiple missense mutations (rs8091003, rs8091117, rs16961689, rs61730306, rs34302455) in DSG1. The DSG1 gene product is a calcium-binding transmembrane glycoprotein component of desmosomes in vertebrate epithelial cells. It connects the cell surface to the keratin cytoskeleton and plays a key role in maintaining epidermal integrity and barrier function[36]. This glycoprotein has been identified as the autoantigen of the autoimmune skin blistering disease pemphigus foliaceus[37] and homozygous

mutations in DSG1 have been showed to result in severe dermatitis, multiple allergies, and metabolic wasting syndrome[36]. A variant downstream of PRKCH (rs2251260, $P = 3.46 \times 10^{-10}$, effect = 0.05) was also significantly associated with yeast infections. None of the variants within 500 kb and in moderate LD ($r^2 > 0.6$) with the index SNP rs2251260 were coding, nor were they reported as an eQTL. However, rs2251260 falls within strong enhancers defined in hepatocellular carcinoma. PRKCH is a member of a family of serine- and threonine-specific protein kinase and is predominantly expressed in epithelial tissues. The PRKCH protein kinase can regulate keratinocyte differentiation[38]. We also found a significant association with a variant in the 14q32.2 gene desert (rs7161578-T, $P = 4.04 \times 10^{-8}$, effect = 0.04). The index SNP is in high LD with many enhancer sequences defined in epidermal keratinocytes. A variant in the same region (rs7152623-A, $P = 3 \times 10^{-15}$, $r^2 = (+)$ 0.98 with rs7161578-T) was implicated in aortic stiffness[39].

**Urinary tract infection**. We found an association between urinary tract infections (UTIs) and variants in PSCA (rs2976388, $P = 3.27 \times 10^{-10}$, effect = 0.04). The index SNP rs2976388 falls within strong enhancers defined in epidermal keratinocytes.

**Table 6 MAGENTA pathway analysis on mumps GWAS**

| Pathway | Pathway P value | Pathway FDR | Gene | Gene P value | SNP | SNP P value |
|---|---|---|---|---|---|---|
| Glycosphingolipid biosynthesis | 1.00E-04 | 0.0029 | FUT1 | 0.00E+00 | rs584768 | 2.38E-62 |
| | | | FUT2 | 0.00E+00 | rs516316 | 9.63E-72 |
| | | | ST3GAL4 | 1.12E-08 | rs3862630 | 1.21E-08 |
| | | | B3GNT1 | 4.20E-02 | rs7947391 | 2.42E-04 |
| | | | GCNT2 | 4.34E-02 | rs147839826 | 7.16E-05 |
| | | | B3GNT2 | 4.46E-02 | rs116508040 | 2.60E-04 |
| | | | ABO | 4.97E-02 | 9:136149709:AC_A | 5.81E-05 |

*CI* confidence interval, *FDR* false discovery rate, *GWAS* genome-wide association study, *HLA* human leukocyte antigen, *MAGENTA* Meta-Analysis Gene-set Enrichment of variaNT Associations, *NA* not applicable, *OR* odds ratio, *SNP* single-nucleotide polymorphism, *UTI* urinary tract infection
The most significant pathway (gene sets) with a gene set enrichment FDR < 0.01 are presented

**Table 7 MAGENTA pathway analysis on tonsillectomy GWAS**

| Pathway | Pathway P value | Pathway FDR | Gene | Gene P value | SNP | SNP P value |
|---|---|---|---|---|---|---|
| Intestinal immune network for IgA production | 2.00E-06 | 3.50E-03 | LTBR | 0.00E + 00 | rs10849448 | 2.35E-35 |
| | | | TNFSF13B | 0.00E + 00 | rs200748895 | 1.20E-17 |
| | | | TNFRSF13B | 0.00E + 00 | rs34557412 | 2.66E-21 |
| | | | CD40 | 3.19E-10 | rs6032664 | 7.31E-12 |
| | | | ICOS | 1.40E-03 | rs231779 | 1.52E-06 |
| | | | CXCR4 | 4.75E-03 | rs2680880 | 6.91E-06 |
| | | | CCL27 | 7.23E-03 | rs12350154 | 6.71E-06 |
| | | | TNFRSF17 | 8.69E-03 | rs456178 | 1.13E-05 |
| | | | IL6 | 9.25E-03 | rs57561814 | 7.45E-06 |
| | | | IL15RA | 1.63E-02 | rs11256464 | 6.99E-06 |
| TACI and BCMA stimulation of B-cell immune responses. | 9.00E-04 | 7.80E-03 | TNFSF13B | 0.00E + 00 | rs200748895 | 1.20E-17 |
| | | | TNFRSF13B | 0.00E + 00 | rs34557412 | 2.66E-21 |
| | | | NFKB1 | 9.08E-14 | rs230523 | 4.54E-14 |
| | | | TNFRSF17 | 8.69E-03 | rs456178 | 1.13E-05 |
| | | | TRAF2 | 1.21E-02 | rs17250694 | 1.56E-05 |
| Ceramide signaling pathway | 3.00E-04 | 7.95E-03 | TNFRSF1A | 0.00E + 00 | rs10849448 | 2.35E-35 |
| | | | NFKB1 | 9.08E-14 | rs230523 | 4.54E-14 |
| | | | MAPK3 | 3.78E-06 | rs12931792 | 4.06E-08 |
| | | | MAP2K2 | 4.75E-04 | rs350906 | 5.34E-07 |
| | | | MYC | 1.13E-03 | rs13279820 | 1.44E-06 |
| | | | TRAF2 | 1.21E-02 | rs17250694 | 1.56E-05 |
| | | | MADD | 3.01E-02 | rs2856661 | 3.76E-05 |
| | | | CTSD | 3.13E-02 | rs7929314 | 3.17E-05 |
| | | | CASP8 | 3.15E-02 | rs190209406 | 3.72E-05 |
| Glucocorticoid receptor regulatory network | 5.00E-04 | 9.93E-03 | NFKB1 | 9.08E-14 | rs230523 | 4.54E-14 |
| | | | MAPK3 | 3.78E-06 | rs12931792 | 4.06E-08 |
| | | | NCOA2 | 9.50E-04 | rs1481046 | 5.65E-07 |
| | | | SELE | 1.66E-03 | rs6691453 | 5.38E-07 |
| | | | BGLAP | 1.94E-03 | rs2075165 | 2.52E-06 |
| | | | IL6 | 9.25E-03 | rs57561814 | 7.45E-06 |
| | | | MAPK11 | 1.20E-02 | rs4076566 | 1.23E-05 |
| | | | SMARCA4 | 1.41E-02 | rs9797720 | 1.58E-05 |
| | | | SPI1 | 2.48E-02 | rs11604680 | 3.38E-05 |
| | | | FKBP5 | 3.06E-02 | rs74853132 | 1.68E-05 |
| | | | ICAM1 | 3.12E-02 | rs74956615 | 4.07E-05 |
| | | | PCK2 | 3.34E-02 | 14:24604399:AAC_A | 5.33E-05 |

*CI* confidence interval, *FDR* false discovery rate, *GWAS* genome-wide association study, *HLA* human leukocyte antigen, *MAGENTA* Meta-Analysis Gene-set Enrichment of variaNT Associations, *NA* not applicable, *OR* odds ratio, *SNP* single-nucleotide polymorphism, *UTI* urinary tract infection
The most significant pathway (gene sets) with a gene set enrichment FDR < 0.01 are presented

It is also in high LD (r2 = 0.95) with rs2294008 ($P = 1.01 \times 10^{-9}$), which is in the 5'-untranslated region promoter sequence of *PSCA* and can cause changes in transcriptional repressor CCCTC-binding factor (CTCF) motif. The *PSCA* gene variants conferred risk of UTI only in females when we further tested the effect in female and male cohorts separately ($P < 1e-11$ in female, and $P > 0.1$ in male). Prior GWASes have found associations between the *PSCA* gene and duodenal ulcer (rs2294008-C, $P = 2 \times 10^{-33}$)[75] and bladder cancer (rs2294008-T, $P = 4 \times 10^{-11}$)[43]. *PSCA* encodes a glycosylphosphatidylinositol-anchored cell membrane glycoprotein of unknown function. This glycoprotein was initially identified as a prostate-specific cell-surface marker[42] and is overexpressed in a large proportion of prostate cancers, but also detected in bladder and pancreatic cancers[43].

**Table 8 MAGENTA pathway analysis on childhood ear infections GWAS**

| Pathway | Pathway P value | Pathway FDR | Gene | Gene P value | SNP | SNP P value |
|---|---|---|---|---|---|---|
| Catalysis of the transfer of organic anions from one side of a membrane to the other | 9.00E-04 | 0.0062 | SLCO1A2 | 1.44E-03 | rs10841784 | 2.76E-06 |
| | | | SLCO1B1 | 3.45E-03 | rs2417971 | 5.67E-06 |
| | | | ABCC3 | 4.74E-03 | rs186049592 | 1.82E-05 |
| | | | SLC22A6 | 3.43E-02 | rs67754977 | 1.20E-04 |
| | | | SLC22A8 | 3.80E-02 | rs67754977 | 1.20E-04 |

*FDR* false discovery rate, *GWAS* genome-wide association study, *MAGENTA* Meta-Analysis Gene-set Enrichment of variaNT Associations, *SNP* single-nucleotide polymorphism
The most significant pathway (gene sets) with a gene set enrichment FDR < 0.01 are presented

We also identified a variant intronic of *FRMD5* (rs146906133, $P = 2.02 \times 10^{-8}$, effect = 0.38) that is in LD with rs138763871 ($r^2 = 0.8$, $P = 1.75 \times 10^{-6}$, effect = 0.33), a missense variant in *STRC* (R1521W). The role of *FRMD5* or *STRC* in UTIs is unclear.

**Tonsillectomy.** Our tonsillectomy GWAS revealed 35 independent GWS associations. We observed strong HLA associations at *HLA-B* in the class I region. The amino acid polymorphism *HLA-B* Val97 ($P = 1.44 \times 10^{-21}$, OR = 1.21) in the peptide-binding cleft of the *HLA-B* protein, accounted for most of the SNP effect (rs41543314, $P = 5.36 \times 10^{-21}$, OR = 1.23, conditional P value is 0.06) and HLA allele effect (*HLA-B\*57:01*, $P = 6.29 \times 10^{-22}$, OR = 1.22, conditional P value is 0.05) in this interval. We also noted additional independent signals in the class II and class III regions. Within *HLA-DRB1* in the class II region, the effect of rs140177540 ($P = 5.02 \times 10^{-15}$, OR = 1.06, conditional P value is 0.001) was largely removed after conditioning on *HLA-DRB1 LeuValGly11* ($P = 1.02 \times 10^{-12}$, OR = 1.06) in the peptide-binding cleft of the *HLA-DRB1* protein.

We detected 34 additional GWS associations (Table 3 and Supplementary Table 4), including signals from genes encoding tumor necrosis factors/receptors superfamily members (*LTBR, TNFRSF13B, TNFSF13B, CD40, TNFAIP3*), myotubularin (*MTMR3*), chemokines (*CXCL13*), mitogen-activated protein kinases/regulators (*IGFBP3, SPRED2, DUSP10, ST5, MAPK3*), Src homology domain-binding proteins (*SBK1, ST5, SH2B3*), disintegrins (*ADAM23, ADAMTS10*), various transcription factors (*NFKB1, HOXA2, FOXA1, IKZF1, MDFIC, TFEC, RERE, TBX1, FOXN1*) and signaling molecules (*WNT2, GNA12, ITSN1*). Many of these genes map to pathways involved in immune and inflammatory processes, while others are important regulators of embryonic development (e.g., most of the transcription factors), and a few are potentially involved in platelet production or hemostasis (*GNA12, SH2B3, RAP1GAP2, ADAM23, ADAMTS10*). When we tested the significant loci for enrichment in canonical pathways, the top over-represented pathway was 'intestinal immune network for IgA production' (Table 7, $P = 2 \times 10^{-6}$ and FDR = 3.5e-3). The genes that contributed significantly in this pathway include *LTBR* (rs10849448, $P = 2.35 \times 10^{-35}$, OR = 1.13), *TNFRSF13B* (rs34557412, missense variant, $P = 2.66 \times 10^{-21}$, OR = 1.59), *TNFSF13B* (rs200748895, $P = 1.20 \times 10^{-17}$, OR = 1.26) and *CD40* (rs6032664, $P = 7.31 \times 10^{-12}$, OR = 1.07). IgA is an antibody that plays a critical role in mucosal immunity. Tonsils belong to nasopharyngeal-associated lymphoid tissues and the generation of B cells is considered one of the major tonsillar functions; secretory dimeric IgA produced by the B cells is capable of preventing absorption and penetration of bacteria and/or viruses into the upper respiratory tract mucosa[44]. Tonsillectomy has been shown to significantly decrease levels of serum IgA and salivary secretory IgA levels[45]. Other identified significant signaling pathways include 'TACI and BCMA stimulation of B-cell immune system' ($P = 9 \times 10^{-4}$ and FDR = 0.008),

'ceramide signaling pathway' ($P = 3 \times 10^{-4}$ and FDR = 0.008) and 'glucocorticoid receptor regulatory network' ($P = 5 \times 10^{-4}$ and FDR = 0.01). The contributing genes are *TNFSF13B* (rs200748895, $P = 1.2 \times 10^{-17}$, OR = 1.26), *TNFRSF13B* (rs34557412, $P = 2.66 \times 10^{-21}$, OR = 1.59), *NFKB1* (rs230523, $P = 4.54 \times 10^{-14}$, OR = 1.07) and *MAPK3* (rs12931792, $P = 4.06 \times 10^{-8}$, OR = 1.06).

Although tonsillectomy has been performed for over 100 years, possible immunological effects of this procedure remain controversial. Many reports have demonstrated that in certain patients tonsillectomy is an effective therapy for psoriasis[34] and rheumatoid arthritis (RA)[46]; the rationale for this effect is unknown. The 'intestinal immune network for IgA production' pathway was also identified as the most significant pathway in a recent GWAS of IgAN[47]. In that study, they also linked the intestinal mucosal inflammatory disorders and inflammatory bowel disease (IBD) with risk of IgAN. Our tonsillectomy GWAS showed significant overlap of the identified loci with those autoimmune and inflammatory disorders[40]. We found five risk loci that are also associated with RA (*HLA, TNFIP3, CD40, SPRED2* and *SH2B3*), four loci associated with IBD (*HLA, MTMR3, TNFAIP3* and *CD40*), four loci associated with ulcerative colitis (*HLA, GNA12, MAPK3* and *NFKB1*), three loci associated with psoriasis (*HLA, SLC12A8* and *1p36.23*), and two loci that are shared with IgAN (*HLA* and *MTMR3*). We observed both concordant and opposing effects compared to these immune-mediated diseases. Our results may help to elucidate the connection between tonsillectomy and these diseases and may provide insight into clinical markers that could be used as indicators of tonsillectomy as a therapy for these diseases.

**Childhood ear infection and myringotomy.** We identified 14 regions that were significantly associated with childhood ear infection. Signals in the HLA region mapped to *HLA-DRB1* in class II region. Our strongest association within *HLA-DRB1* was with *HLA-DRB1 Gln96* in the peptide-binding groove. After conditioning on *HLA-DRB1 Gln96* ($P = 2.11 \times 10^{-12}$, OR = 0.92), the effects of rs4329147 ($P = 9.55 \times 10^{-12}$, OR = 1.11, conditional P value is 0.27) and *DQB1\*06:02* ($P = 5.79 \times 10^{-10}$, OR = 0.92, conditional P value is 0.85) were eliminated.

We found 13 additional GWS signals (Table 2 and Supplementary Table 5). Although MAGENTA pathway analysis did not identify significant pathways (Table 8) of interest, variants in *FUT2* and *ABO*, which are involved in the GSL biosynthesis pathway and were implicated in our mumps GWAS, were also significantly associated with childhood ear infection. The risk allele rs681343(C) ($P = 3.51 \times 10^{-30}$, OR = 1.11) is a synonymous mutation in *FUT2* and is in almost complete LD with rs601338 (G) ($r^2 = 0.9993$), the 'secretor' (se) allele that is associated with higher risk of childhood ear infection in our data. This is consistent with a previous report in which secretors were over-represented among patients with upper respiratory infections[18].

Our second most significant association was with a variant in *TBX1* (rs1978060, $P = 1.17 \times 10^{-19}$, OR = 1.09). The low-risk allele rs1978060 (A) is in LD ($r^2 = 0.45$) with rs72646967 (C), a missense (N397H) mutation in *TBX1* that was also implicated in our tonsillectomy GWAS. T-box genes encode transcription factors that play important roles in tissue and organ formation during embryonic development. In mice, *TBX1* haplo-insufficiency in the DiGeorge syndrome region has been showed to disrupt the development of the pharyngeal arch arteries[41] and the middle and outer ear[59]. We also identified other genes involved in developmental processes, including *MKX* (rs2808290, $P = 5.09 \times 10^{-16}$, OR = 1.07), *FGF3* (rs72931768, $P = 2.63 \times 10^{-9}$, OR = 1.09) and *AUTS2* (rs35213789, $P = 3.75 \times 10^{-9}$, OR = 1.06). The *MKX* product is an IRX family-related homeobox transcription factor and has been shown to play a critical role in tendon differentiation by regulating type I collagen production in tendon cells[49]. *FGF3* is a member of the fibroblast growth factor family, which is involved in embryonic development, cell growth, morphogenesis, tissue repair and invasion. A study of similar genes in mouse and chicken suggested a role in inner ear formation[50]. *AUTS2* has been implicated in neurodevelopment and is a candidate gene for autism spectrum disorders and developmental delay[76]. Finally, we identified missense mutations in *CDHR3* (rs114947103, $P = 5.40 \times 10^{-9}$, OR = 1.07) and *PLG* (rs73015965, $P = 3.78 \times 10^{-8}$, OR = 1.43). The index SNP rs114947103 is in almost complete LD with rs6967330 ($r2 = 0.97$), a missense variant in *CDHR3*, that was recently identified in a GWAS as a susceptibility locus for asthma[52]. The biological role of *CDHR3* is not clear, but it is highly expressed in airway epithelium and belongs to the cadherin family that is involved in cell adhesion and epithelial polarity[52]. Mutations in the *PLG* gene could cause congenital plasminogen deficiency, which results in inflamed growths on the mucous membranes. Studies in mice have showed that *PLG* plays an essential role in protecting against the spontaneous development of chronic otitis media (middle ear infection) and have also suggested the possibility of using *PLG* for clinical therapy of certain types of otitis media[53].

The same variant in *TBX1* that was associated with tonsillectomy and childhood ear infection (see earlier) was also found to be significantly associated with myringotomy (rs1978060, $P = 3.34 \times 10^{-10}$, OR = 1.17).

## Discussion

This is the first description of GWASes for numerous common infectious diseases in a single study. Our GWASes replicated the previously reported associations with shingles and tonsillectomy at much higher significance (Supplemental Table 1). In other instances our GWASes did not replicate prior findings and instead identified novel associations. Our GWASes of HBV infection, mononucleosis and positive TB test identified significant HLA associations; however, they are not in strong LD with any of the previously reported loci (Supplemental Table 1). For the associations that did not replicate, several factors may have reduced our power. First, there was genetic heterogeneity in different ethnic groups due to differences in allele frequency and LD structure, especially in the HLA region. Many complex diseases have been shown to be influenced by different genetic determinants in different ethnic groups[54]. Second, there were differences in how the disease phenotypes were determined. Our hepatitis B phenotype was defined by participants' self-declared history of HBV infection and thus combines cases of subclinical infection that were detected after the fact, acute infections that spontaneously cleared, successfully treated infections, and chronic persistent infection cases ('Survey and phenotype scoring logic' in

Supplementary Notes 1). This definition differs from studies that focus solely on chronic HBV carriers. Similarly, our GWAS of patients with a history of a positive TB tests tested a different phenotype than published studies of patients with active TB.

Insufficient power due to small sample size may have in some instances contributed to our results differing from previous studies. Our GWASes of susceptibility to meningitis (based on a sample of 842 cases) failed to replicate the findings at *CFH*, but identified a significant association at *CA10*. Moreover, our case definition of self-reported 'meningitis' may be confounded by the fact that bacterial meningitis is frequently considered in patients with presenting to the hospital with severe headache and fever, a definitive diagnosis is made in a minority of cases[55]. In other words, patients initially suspected to have bacterial meningitis may ultimately be diagnosed with other syndromes such as viral meningitis, stroke, subdural empyema and cerebral abscess[55].

Notwithstanding the limitations, our study identified numerous significant associations and also suggested crucial roles of innate immune responses in protective immunity to multiple common infections. For example, two genes involved in innate immunity of skin, *LCE3E* and *DSG1*, were significantly associated with plantar warts and yeast infections, respectively. We identified the greatest number of independent associations for tonsillectomy ($n = 35$) and childhood ear infection ($n = 14$), two relatively nonspecific phenotypes. The large number of associations may reflect, in part, the large sample sizes for these two traits ($n = 173,412$ and $121,810$, respectively). There were many associations between both tonsillitis and middle ear infections with genes playing roles in the innate immune response. Lymphoid hyper-reaction throughout the mucosa-associated lymphoid tissue system has been suggested as a molecular mechanism underlying the genetic association with tonsillectomy[9]. In addition, we discovered associations between both syndromes and genes with roles in embryonic development, underscoring the important role of host anatomy in both syndromes. Patients undergo tonsillectomy for multiple reasons, including recurrent tonsillitis, obstructive sleep apnea and nasal airway obstruction[56]. Similarly, childhood ear infections are strongly influenced by the shape of the Eustachian tube[57]. The missense mutation (rs72646967-C, N397H) in *TBX1*, which is essential for inner ear development[58, 59], is associated with a lower risk of both tonsillectomy and inner ear infections (Supplementary Fig. 5B).

In the HLA region, we found that viral diseases—e.g., chickenpox, shingles, cold sores, mononucleosis (all caused by members of the human herpesvirus) as well as mumps (caused by mumps virus)—were mainly associated with variation in class I molecules (Fig. 1). The bacterial diseases—specifically having a positive TB test (caused by *Mycobacterium tuberculosis*), scarlet fever (caused by *Streptococcus pyogenes*), and childhood ear infection (primarily caused by bacteria)—were mainly associated with variation in class II molecules. Tonsillectomy and pneumonia, caused by either bacteria or viruses, were associated with both class I and class II molecules. These observations are consistent with previous knowledge about antigen presentation; viruses mostly replicate within nucleated cells in the cytosol and produce endogenous antigens that are presented by class I major histocompatibility complex (MHC) molecules, while—with notable exceptions—the bacteria responsible for the diseases in question grow primarily extracellularly and are taken up by endosomal compartments where they are processed for presentation by class II MHC molecules[60]. The two intracellular pathways of protein processing may not be completely separate. Activation of CD8 + , HLA class I-restricted T cells by exogenous antigens has been reported[61] and HLA class II-restricted cytotoxic T cells that recognize endogenously synthesized antigen, such as HBV envelope antigens, have also been described[62].

Some diseases, however, do not follow the common principles of antigen presentation. HPV infections, causing plantar warts, are exclusively intraepithelial and there is no viremia or cytolysis in the infection cycle. The primary response to HPV antigens is more likely to be initiated by the antigen-presenting cells of squamous epithelia, the Langerhans cells (LCs). LCs capture antigens by macropinocytosis and receptor-mediated endocytosis, and then initiate class II processing of exogenous antigens[48], which may explain why our plantar warts GWAS mainly identified associations with MHC class II molecules. In other cases, our results suggest even more complex interactions. Strep throat (caused by *S. pyogenes*) has both MHC class I and class II associations. Some of our reported strep throat cases may represent secondary bacterial infections following a viral upper respiratory infection, meaning that the discovered MHC class I associations actually reflect the initiating viral infection[63]. Some cases may have been misdiagnosed or misreported as strep throat, given that many sore throats are actually caused by viruses[63]. A final possibility is that HLA class I molecules can bind bacterial peptides derived from exogenous proteins that are internalized by endocytosis or phagocytosis, a process called cross-presentation[64].

Although additional studies will be required to validate our associations, our findings are an important step toward dissecting the host genetic contribution to variation in response to infections. Research insights into infectious diseases will help to drive new diagnostic approaches and perhaps new therapies and preventions. Moreover, our results may also yield insights into autoimmune disorders that are associated with infectious triggers. A postulated mechanism for the relationship between infection and immunological disorders is that certain pathogens elicit immune responses to cross-reactive self epitopes, thereby initiating a cycle of damage and further immune activation[65]. Indeed, our study identified at least 10 infectious disease associations with genes that have been previously associated with autoimmune diseases.

## Methods

**Subjects.** Participants used in our GWASes were drawn from the customer base of 23andMe, Inc, a personal genetics company. All individuals included in the analyses provided informed consent and answered surveys online according to our human subjects protocol, which was reviewed and approved by Ethical and Independent Review Services, a private institutional review board http://www.eandireview.com). Over 200,000 genotyped participants were included in analyses based on selection for having >97% European ancestry as determined through an analysis of local ancestry[51] and completing surveys about their history of common infections (Table 1, Supplementary Table 2, 'Survey and phenotype scoring logic' in Supplementary Notes 1). Except for the history of chickenpox infections, participants were also asked whether they have ever received the chickenpox vaccine, so that vaccinated individuals were excluded from the controls in the chickenpox study. The vaccination status for many other infections such as mumps, measles and rubella was not surveyed and the controls for them included vaccinated individuals ('Surveys and phenotype scoring logic' in Supplementary Note 1). For each study, a maximum set of unrelated individuals was chosen using a segmental identity-by-descent estimation algorithm[66]. Individuals were defined as related if they shared more than 700 cM of identity-by-descent, including regions where the two individuals share either one or both genomic segments identity-by-descent. This level of relatedness (involving ~ 20% of the human genome) corresponds approximately to the minimal expected sharing between first cousins in an outbred population.

**Genotyping and SNP imputation.** DNA extraction and genotyping were performed on saliva samples by CLIA-certified and CAP-accredited clinical laboratories of Laboratory Corporation of America. Samples were genotyped on one of four genotyping platforms. The V1 and V2 platforms were variants of the Illumina HumanHap550 + BeadChip and contained a total of about 560,000 SNPs, including about 25,000 custom SNPs selected by 23andMe. The V3 platform was based on the Illumina OmniExpress + BeadChip and contained a total of about 950,000 SNPs and custom content to improve the overlap with our V2 array. The V4 platform in current use is a fully custom array of about 950,000 SNPs and includes a lower redundancy subset of V2 and V3 SNPs with additional coverage of

lower-frequency coding variation. Samples that failed to reach 98.5% call rate were re-analyzed. Individuals whose analyses failed repeatedly were re-contacted by 23andMe customer service to provide additional samples, as is done for all 23andMe customers.

Participant genotype data were imputed using the March 2012 'v3' release of the 1000 Genomes Project reference haplotypes[67]. We phased and imputed data for each genotyping platform separately. First, we used BEAGLE[68] (version 3.3.1) to phase batches of 8000–9000 individuals across chromosomal segments of no more than 10,000 genotyped SNPs, with overlaps of 200 SNPs. We excluded SNPs with minor allele frequency < 0.001, Hardy–Weinberg equilibrium $P < 1 \times 10^{-20}$, call rate <95%, or large allele frequency discrepancies compared to the European 1000 Genomes Project reference data. Frequency discrepancies were identified by computing a $2 \times 2$ table of allele counts for European 1000 Genomes samples and 2000 randomly sampled 23andMe customers with European ancestry, and identifying SNPs with a $\chi^2$ $P < 10^{-15}$. We imputed each phased segment against all-ethnicity 1000 Genomes haplotypes (excluding monomorphic and singleton sites) using Minimac2[69] with five rounds and 200 states for parameter estimation.

For the non-pseudoautosomal region of the X chromosome, males and females were phased together in segments, treating the males as already phased; the pseudoautosomal regions were phased separately. We then imputed males and females together using Minimac as with the autosomes, treating males as homozygous pseudo-diploids for the non-pseudoautosomal region.

**GWAS analysis.** For case control comparisons, we tested for association using logistic regression, assuming additive allelic effects. For quantitative traits, association tests were performed using linear regression. For tests using imputed data, we use the imputed dosages rather than best-guess genotypes. We included covariates for age, gender, and the top five principal components to account for residual population structure. The association test $P$ value was computed using a likelihood ratio test, which in our experience is better behaved than a Wald test on the regression coefficient. Results for the X chromosome were computed similarly, with men coded as if they were homozygous diploid for the observed allele.

**SNP function annotation.** To explore whether any of the significant SNPs identified might link to functional mutations or have potential regulatory functions, we used the online tools HaploReg (http://www.broadinstitute.org/mammals/haploreg/haploreg.php) to confirm the location of each SNP in relation to annotated protein-coding genes and/or non-coding regulatory elements. We queried only the variants that were within 500 kb of and in moderate LD ($r^2 > 0.6$) with the index SNP.

**Imputation of HLA classical alleles and respective amino acid variations.** HLA imputation was performed with HIBAG[70], an attribute bagging based statistical method that comes as a freely available R package and includes a pre-fit classifier. This classifier was trained from a large database of individuals with known HLA alleles and SNP variation within the HLA region. Over 98% of the tagging SNPs used in HIBAG were genotyped and passed quality control on 23andMe's platform. We imputed allelic dosage for HLA-A, B, C, DPB1, DQA1, DQB1 and DRB1 loci at four-digit resolution. We used the default settings of HIBAG and the run time for 100,000 samples was about 10 h on our cluster.

Using an approach suggested in [71], we downloaded the files that map HLA alleles to amino acid sequences from https://www.broadinstitute.org/mpg/snp2hla/ and mapped our imputed HLA alleles at four-digit resolution to the corresponding amino acid sequences; in this way we translated the imputed HLA allelic dosages directly to amino acid dosages. We encoded all amino acid variants in the 23andMe European samples as biallelic markers, which facilitated downstream analysis. For example, position 45 of HLA-B protein had five different alleles (E: Glu, M: Met, T: Thr, K: Lys, G: Gly), we first encoded the position using five binary markers, each corresponding to the presence or absence of each allele (e.g., *HLA-B Gly45* tags Gly at position 45 of HLA-B protein). For positions having three or more alleles, we also created markers that tag multiple alleles, each corresponding to the presence or absence of the multiple alleles (e.g., *HLA-B ThrGly45* tags Thr and Gly at position 45 of *HLA-B* protein). Thus, we created binary indicators for all possible combinations of amino acid variants. We use this naming convention for amino acid polymorphisms throughout this paper.

We imputed 292 classical HLA alleles at four-digit resolution and 2395 biallelic amino acid polymorphisms. Similar to the SNP imputation, we measured imputation quality using $r^2$, which is the ratio of the empirically observed variance of the allele dosage to the expected variance assuming Hardy–Weinberg equilibrium. The imputation quality ($r^2$) of the top associated HLA alleles and amino acids are in Supplementary Table 8.

**HLA region fine mapping.** To test associations between imputed HLA allele/amino acid dosages and phenotypes, we performed logistic or linear regression using the same set of covariates used in the SNP-based GWAS for that phenotype. We applied a forward stepwise strategy, within each type of variant, to establish statistically independent signals in the HLA region. Within each variant type (e.g., SNP, HLA allele and HLA amino acid), we first identified the most

strongly associated signals (lowest *P* value) for each disease and performed forward iterative conditional regression to identify other independent signals if the conditional *P* value was $< 5 \times 10^{-8}$. All analyses controlled for sex and five principal components of genetic ancestry. The *P* values were calculated using a likelihood ratio test. The iterative conditional regression dissected HLA signal into independent HLA associations. Within each identified HLA locus (HLA-A, B, C, DPB1, DQA1, DQB1 and DRB1), we further carried out reciprocal analyses, which are the conditional analyses across variants types, to see if the association can be attributed to the amino acid polymorphism within each HLA locus.

**Pathway analysis**. To better understand how multiple genes in the same pathway may contribute to certain infections, we performed pathway analysis using MAGENTA[72], which tests for enrichment of genetic associations in predefined biological processes or sets of functionally related genes, using GWAS results as input. We used gene sets of 1320 canonical pathways from the Molecular Signatures Database (MsigDB) compiled by domain experts[73] and default settings of MAGENTA tool. We reported the most significant pathway (gene sets) with a gene set enrichment FDR < 0.01.

**Data availability**. The GWAS summary statistics for the top 8000 SNPs for each phenotype are available to download in Supplementary Data 1. The association test statistics for the HLA alleles and HLA amino acid with *P* value < 0.05 for each phenotype are available to download in Supplementary Data 2. Researchers can request to access the full set of GWAS statistics by applying to 23andMe research collaboration program.

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

# ARTICLE

47. Gharavi, A. G. et al. Genome-wide association study identifies susceptibility loci for IgA nephropathy. *Nat. Genet.* **43**, 321–327 (2011).
48. Stanley, M. Immune responses to human papillomavirus. *Vaccine* **24**, S16–S22 (2006).
49. Ito, Y. et al. The Mohawk homeobox gene is a critical regulator of tendon differentiation. *Proc. Natl Acad. Sci. USA* **107**, 10538–10542 (2010).
50. Sensi, A. et al. LAMM syndrome with middle ear dysplasia associated with compound heterozygosity for FGF3 mutations. *Am. J. Med. Genet. A.* **155A**, 1096–1101 (2011).
51. Durand, E. Y., Do, C. B., Mountain, J. L. & Macpherson, J. M. Ancestry composition: a novel, efficient pipeline for ancestry deconvolution. Preprint at http://biorxiv.org/content/early/2014/10/18/010512 (2014).
52. Bønnelykke, K. et al. A genome-wide association study identifies CDHR3 as a susceptibility locus for early childhood asthma with severe exacerbations. *Nat. Genet.* **46**, 51–55 (2014).
53. Eriksson, P.-O., Li, J., Ny, T. & Hellström, S. Spontaneous development of otitis media in plasminogen-deficient mice. *Int. J. Med. Microbiol.* **296**, 501–509 (2006).
54. Burgner, D., Jamieson, S. E. & Blackwell, J. M. Genetic susceptibility to infectious diseases: big is beautiful, but will bigger even be better? *Lancet Infect. Dis.* **6**, 653–663 (2006).
55. Brouwer, M. C. & van de Beek, D. Earlier treatment and improved outcome in adult bacterial meningitis following guideline revision promoting prompt lumbar puncture. *Clin. Infect. Dis.* **61**, 664–665 (2015).
56. Baugh, R. F. et al. Clinical practice guideline: tonsillectomy in children. *Otolaryngol. Head Neck Surg.* **144**, S1–S30 (2011).
57. Lim, D. J. Functional morphology of the mucosa of the middle ear and Eustachian tube. *Ann. Otol. Rhinol. Laryngol.* **85**, 36–43 (1976).
58. Raft, S., Nowotschin, S., Liao, J. & Morrow, B. E. Suppression of neural fate and control of inner ear morphogenesis by Tbx1. *Development* **131**, 1801–1812 (2004).
59. Vitelli, F. et al. TBX1 is required for inner ear morphogenesis. *Hum. Mol. Genet.* **12**, 2041–2048 (2003).
60. Blum, J. S., Wearsch, P. A. & Cresswell, P. Pathways of antigen processing. *Annu. Rev. Immunol.* **31**, 443–473 (2013).
61. Rock, K. L., Gamble, S. & Rothstein, L. Presentation of exogenous antigen with class I major histocompatibility complex molecules. *Science* **249**, 918–921 (1990).
62. Penna, A. et al. Hepatitis B virus (HBV)-specific cytotoxic T-cell (CTL) response in humans: characterization of HLA class II-restricted CTLs that recognize endogenously synthesized HBV envelope antigens. *J. Virol.* **66**, 1193–1198 (1992).
63. Carapetis, J. R., McDonald, M. & Wilson, N. J. Acute rheumatic fever. *Lancet* **366**, 155–168 (2005).
64. Rock, K. L. & Shen, L. Cross-presentation: underlying mechanisms and role in immune surveillance. *Immunol. Rev.* **207**, 166–183 (2005).
65. Ercolini, A. M. & Miller, S. D. The role of infections in autoimmune disease. *Clin. Exp. Immunol.* **155**, 1–15 (2009).
66. Henn, B. M. et al. Cryptic distant relatives are common in both isolated and cosmopolitan genetic samples. *PLoS ONE* **7**, e34267 (2012).
67. 1000 Genomes Project Consortium. A map of human genome variation from population-scale sequencing. *Nature* **467**, 1061–1073 (2010).
68. Browning, S. R. & Browning, B. L. Rapid and accurate haplotype phasing and missing-data inference for whole-genome association studies by use of localized haplotype clustering. *Am. J. Hum. Genet.* **81**, 1084–1097 (2007).
69. Fuchsberger, C., Abecasis, G. R. & Hinds, D. A. minimac2: faster genotype imputation. *Bioinformatics* **31**, 782–784 (2014).
70. Zheng, X. et al. HIBAG--HLA genotype imputation with attribute bagging. *Pharmacogenomics J.* **14**, 192–200 (2014).
71. Jia, X. et al. Imputing amino acid polymorphisms in human leukocyte antigens. *PLoS ONE* **8**, e64683 (2013).
72. Segrè, A. V. et al. Common inherited variation in mitochondrial genes is not enriched for associations with type 2 diabetes or related glycemic traits. *PLoS Genet.* **6**, e1001058 (2010).
73. Subramanian, A. et al. Gene set enrichment analysis: a knowledge-based approach for interpreting genome-wide expression profiles. *Proc. Natl Acad. Sci. USA* **102**, 15545–15550 (2005).
74. McGovern, D. P. B. et al. Fucosyltransferase 2 (FUT2) non-secretor status is associated with Crohn's disease. *Hum. Mol. Genet.* **19**, 3468–3476 (2010).
75. Tanikawa, C. et al. A genome-wide association study identifies two susceptibility loci for duodenal ulcer in the Japanese population. *Nat. Genet.* **44**, 430–434 S1–2 (2012).
76. Jolley, A. et al. De novo intragenic deletion of the autism susceptibility candidate 2 (AUTS2) gene in a patient with developmental delay: a case report and literature review. *Am. J. Med. Genet. A* **161A**, 1508–1512 (2013).

## Acknowledgements

We thank the customers of 23andMe who consented to participate in the studies and provided answers to survey questions. We also thank the employees of 23andMe, who together have made this research possible. We thank 23andMe research team members, including Michelle Agee, Babak Alipanahi, Adam Auton, Robert K. Bell, Katarzyna Bryc, Sarah L. Elson, Pierre Fontanillas, Nicholas A. Furlotte, Karen E. Huber, Aaron Kleinman, Nadia K. Litterman, Matthew H. McIntyre, Joanna L. Mountain, Carrie A.M. Northover, Steven J. Pitts, J. Fah Sathirapongsasuti, Olga V. Sazonova, Janie F. Shelton, Suyash Shringarpure, Vladimir Vacic, and Catherine H. Wilson, who build our research analytical pipeline. We specially thank 23andMe's Pierre Fontanillas for discussion on the heritability estimation. S.M.N. is supported by the US National Institutes of Health (grant R01 AI108992), an Investigator in the Pathogenesis of Infectious Disease award from the Burroughs Wellcome Fund, and a Scholar in the Biomedical Sciences award from the Pew Charitable Trusts.

## Author contributions

C.T. and D.A.H. analyzed the data and wrote the manuscript. B.S. and S.M.N. helped interpret the results and revise the manuscript. A.K. and J.Y.T. designed the study. N.E. developed analytical tools. 23andMe research Team built the analytical pipeline.

## Additional information

**Competing interests:** C.T., D.A.H., B.S.H., A.K.K., N.E. and J.Y.T. were/are employed by 23andMe, and own stock or stock options in 23andMe. The remaining author declares no competing financial interests.

