## [Peer Review File · Nature Communications]

Reviewers' comments:

Reviewer #1 (Remarks to the Author):

This is a significant study for the genetics of ID spanning 23 disease conditions with 200,000 subjects. For many of these diseases this is the first published GWAS. Strengths of the study include its massive size and solid GWAS analysis including imputation for classical HLA alleles and conditional analysis to deconvolute HLA signals. GWA analysis appears to have been done appropriately on individuals of European descent with phasing and imputation based on 1000 Genomes and association by logistic or linear regression with covariates of age, gender, top 5 PCs. However, the authors carry out little meta-analysis to more fully characterize their unique dataset except for pairwise LD score correlation revealing several expected positive correlations. Therefore, the GWAS findings are of interest to those in ID genetics, especially those working on the specific infections.

The authors report several HLA associations of high significance, many for the first time with particular IDs that have not undergone GWAS. Beyond HLA, the authors also make some new discoveries that may eventually lead to more biological insight. The suggestive IFNL4 association with HepA warrants additional followup studies (beyond the scope of this manuscript), as does the LCE3E association with plantar warts, and the DSG1 association with candidiasis. Thus the overall biological significance of this paper is somewhat limited currently, but I suspect it will serve to spur additional work to discover how these new variants impact disease pathogenesis.

However, there are several major and minor issues that somewhat dampen my enthusiasm:

Major issues that must be addressed:

1) Examining Supplemental Table 2 it is clear that there were some fairly robust negative genetic correlations detected; for example, shingles and hepatitis A $r_g = -0.62$ and $p = 0.02$. How do they interpret this? It is unclear why the authors set negative genetic correlations to 0 in the figure. The justification that all negative genetic correlations were not statistically significant does not appear to be true and even if true, the decision to set all these values to 0 seems strange. Are all positive correlations significant on the chart?

2) Heritabilities are not commented on in the manuscript which seems odd as some heritability is a requirement for a GWAS approach to work. However, heritability based on the LD score method is provided in Table S2. Surprisingly, the values are miniscule for most of the traits, including as low as 0.0036 for hepatitis B and never ranging higher than 5%. The authors must address the following points in regards to this: 1) Are such low heritabilities compatible with the GWAS findings? Do they explain more than 100% of estimated heritability for some traits? 2) How do these values compare with published estimates from more traditional methods for these same diseases? 3) LD score is a relatively new method and these very low heritabilities warrant estimation of heritability using a different method, such as the GREML method implemented in GCTA. This will help ensure that these estimates are not somehow erroneous either due to some user error or bug in the LD score software.

3) Greater detail should be given describing how cases and controls were defined for each study in the Methods. Why do the total number of individuals for most of the studies add up to much less than 200,000? How does exposure to vaccine (for chicken pox, measles, mumps, rubella, etc...) effect inclusion and classification in the study? ...I now see that the survey and logic for selecting cases and controls is in the Supplemental information. While this is good, as case/control definitions are central to what's actually being tested here, the overall logic should be described in the text and the supplemental referenced.

4) No odds ratios are reported in the text, so effect size of these SNPs are unable to be gauged. I realize they are listed in table 2, but it is helpful not to have to look up the ones that are discussed

in evaluating the possible significance of the results. Therefore, ORs should be provided in the same instances where p-values are given in the text. The authors might also want to indicate that the ORs they have found in many instances are quite large compared to most GWAS findings to further emphasize the significance of their findings.

5) No information is given on how the actual GWAS data will be disseminated. The results of these GWAS should be made publicly available upon publication. This should at least include all rsIDs, risk alleles, p-values, and ORs for each of the 23 studies either with the online manuscript or a publicly accessible hosted webpage. In this way, identified SNPs that did not reach genome-wide significance can be further examined and other investigators can perform additional meta-analyses on this extraordinarily rich dataset. I imagine these studies could become a frequently cited resource if the data are made available in this manner.

6) QQ plots should be provided to demonstrate lack of overall inflation of test statistics

Suggestions:

1) It would be informative to more comprehensively catalog how well previous GWAS of the same infectious diseases did or did not show replication. This is an opportunity to demonstrate how well replication occurs for "genome-wide significant" SNPs across several different IDs. This analysis could inform how well replication can be expected to occur and how much of a problem heterogeneity between study populations may be.

2) The GWA results for tonsillectomy were particularly interesting and the overlap with autoimmune diseases thought-provoking. If the authors wished, they could quantify the significance of the overlap and level of enrichment with autoimmune diseases but this is up to the discretion of the authors.

3) For variants that come up repeatedly in ID, such as FUT2, a diagram that depicts the directionality of this variant for risk of various diseases would be helpful and possible mechanism for this directionality could be discussed.

Minor comments:

1) Typos of p20—extra period and mistakenly italicized text

2) P23—the authors state that "bacteria grow extracellularly" but this is not strictly true. This is certainly true of streptococcus and many other bacteria, but TB for example can grow intracellularly within macrophages, while other bacteria such as Chlamydia are obligate intracellular pathogens.

Overall, I enjoyed reading this manuscript and feel the authors have put together a nice study that will be of high interest to the ID genetics community. However, the issues listed above do substantially detract from what could be an even better manuscript and need to be corrected.

Reviewer #2 (Remarks to the Author):

This is a very large GWAS on self-reported infectious disease phenotypes for more than 200,000 European research participants. The analysis is a classic GWAS complemented by a more detailed investigation of the HLA regions because of the expected concentration of hits there.

The strength of the study is in the numbers. The weakness of the study is in the interpretation of the data (some examples below). Suboptimal interpretation occurs both in the area of infectious diseases, and also in the area of immunogenetics. It would be highly recommended that the authors would join to the study respected researchers and clinicians in those areas. This would dramatically increase the value of the report.

Specific comments and examples:

1. "There was a phenotypic correlation (Pearson $r = 0.41$, $P < 2.2 \times 10^{-16}$) among women who reported both UTI and yeast infections in our cohort. These two infections can have similar symptoms and both cause discomfort in vaginal area, but different infectious agents cause them." Comment: in reality, the correlation is established by the triggering of the candidae infection by the antibiotics used to treat the bacterial infection. There less a shared pathogenesis and genetics, than a secondary effect of treatment.

2. Tonsillectomy. The complex pattern of association with innate and acquired immunity loci is interpreted as reflecting diversity in pathogens. Heterogeneity of pathogens and mechanisms would dilute the signal – contrary to what is described. More plausibly, the results reflect the uniqueness of inflammatory response (ie., exaggerated) that may accompany the response to infectious insults.

3. Hepatitis A. The study maps (or not, according to the statistical threshold) this infection to IL28B. This is the locus of Hepatitis C. The most parsimonious explanation is that the responders are attributing their "hepatitis" experience to hepatitis A (a picornavirus) instead as to Hepatitis C (a flavivirus).

4. The manuscript is quite successful at trying to tease out the signals in the HLA regions. Mapping, conditional analyses, imputation of groove amino acids are all well founded steps. And somehow, at the end the reader feels that the data is not well represented, well presented graphically (showing the sharing and proximity of signals across the regions for all pathogens). The lack of solution of signals in the HCP5 regions is also a surprise, because their conditioning analysis should have supported the tagging of HLA alleles – but it is described as the contrary: the HLA regions loses statistical power when conditioned by the HCP5 alleles.

5. GWAS standards. Would not report alleles that are labelled as "moderately associated" (eg. DQB1*06:02 was moderately associated with hepatitis B ($P = 5.47 \times 10^{-6}$)). Similarly, there is some contradiction on the reports on Hepatitis A: Table 1 lists it as "no-GWAS" hit, but it is discussed in the text, and presented as positive association in Table 2

6. GWAS standards. While the study is announced as "European" population, and correctly using PC1-PC5 axes for population stratification control, there is a description of the study including 3% non-EUR. It may be desirable to drop the minor population of non-EUR to avoid some uncorrected structure and different epidemiology.

7. GWAS standards. Figure 1. Summary of Independent HLA Signals – Unclear multiple testing correction. This is also true at other sections in the manuscript. There should be a clear statement on the study power threshold and its strict application.

8. Order of the text: The initial results describe the pairwise LD score correlation on GWAS statistics. This should come at the end – once that the result of the study have been presented. Otherwise it is confusing.

9. Discussion. There is extensive repetition of results. The text should be used to highlight the high level understanding of the broad mapping across the HLA regions, and the obvious associations across the innate immunity. In addition, there should be a better discussion of previous GWAS studies. There is minimal effort at summarizing and contrasting the existing literature on infectious diseases GWAS. For example, meningococcal meningitis – common cause of meningitis, is associated with complement deficiency. Here the broader "bacterial meningitis" is associated with CA10 (carbonic anhydrase X). It may be correct – but deserves some reflection on the discrepancy – and a report on the p values for the alleles that have been previously reported.

Minor comments:

1. Fig S3 - The purple dotted line corresponds to $P=5 \times 10^{-8}$, - It does not appear to match the plot
2. "A-Val95" means amino acid Val at position 95 of the HLA-A protein. We use this naming convention throughout the paper, which is composed from the name of the HLA protein (A), followed by the three-letter code for the amino acid (Val) and its location in the protein". This non-conventional nomenclature is ill-advised. It may be interpreted as Ala95Val.
3. Incomplete references:
 36. Learn the Signs and Symptoms of TB Disease.
 45. Pneumococcal Disease.
 48. Medical microbiology. (University of Texas Medical Branch at Galveston, 1996).
 78. Ear Infections in Children.
 89. Sore Throat.

Reviewer #1 (Remarks to the Author):

This is a significant study for the genetics of ID spanning 23 disease conditions with 200,000 subjects. For many of these diseases this is the first published GWAS. Strengths of the study include its massive size and solid GWAS analysis including imputation for classical HLA alleles and conditional analysis to deconvolute HLA signals. GWA analysis appears to have been done appropriately on individuals of European descent with phasing and imputation based on 1000 Genomes and association by logistic or linear regression with covariates of age, gender, top 5 PCs. However, the authors carry out little meta-analysis to more fully characterize their unique dataset except for pairwise LD score correlation revealing several expected positive correlations. Therefore, the GWAS findings are of interest to those in ID genetics, especially those working on the specific infections.

The authors report several HLA associations of high significance, many for the first time with particular IDs that have not undergone GWAS. Beyond HLA, the authors also make some new discoveries that may eventually lead to more biological insight. The suggestive IFNL4 association with HepA warrants additional followup studies (beyond the scope of this manuscript), as does the LCE3E association with plantar warts, and the DSG1 association with candidiasis. Thus the overall biological significance of this paper is somewhat limited currently, but I suspect it will serve to spur additional work to discover how these new variants impact disease pathogenesis.

However, there are several major and minor issues that somewhat dampen my enthusiasm:

Major issues that must be addressed:

1) Examining Supplemental Table 2 it is clear that there were some fairly robust negative genetic correlations detected; for example, shingles and hepatitis A $r_g = -0.62$ and $p = 0.02$. How do they interpret this? It is unclear why the authors set negative genetic correlations to 0 in the figure. The justification that all negative genetic correlations were not statistically significant does not appear to be true and even if true, the decision to set all these values to 0 seems strange. Are all positive correlations significant on the chart?

Response: Thanks for pointing it out. It turns out that SNP imputation quality is a significant confounder for LD score regression. To prevent the bias from imputation quality, we re-calculated the genetic correlation after removing poorly-imputed SNPs (as defined by imputation quality $r^2 < 0.9$). There is no significant negative correlation in the new result (Supplementary Figure 4 and Supplementary Table 6). The shingles and hepatitis_A now has $r_g = -0.42$ but $P = 0.3$, which is not significant.

We also modified the heat map plot (Supplementary Figure 4) with 'Blue Square' representing positive genetic correlation; 'Red square' representing negative genetic correlation. Darker colors represent larger genetic correlations. Larger squares correspond to more significant P values; genetic correlations that are significantly different from zero after Bonferroni correction for the ~250 tests (p-value < 2e-4) in this correlation analysis are shown as full-sized squares.

2) Heritabilities are not commented on in the manuscript which seems odd as some heritability is a requirement for a GWAS approach to work. However, heritability based on the LD score method is provided in Table S2. Surprisingly, the values are miniscule for most of the traits, including as low as 0.0036 for hepatitis B and never ranging higher than 5%. The authors must address the following points in regards to this: 1) Are such low heritabilities compatible with the GWAS findings? Do they explain more than 100% of estimated heritability for some traits? 2) How do these values compare with published estimates from more traditional methods for these same diseases? 3) LD score is a relatively new method and these very low heritability warrant estimation of heritability using a different method, such as the GREML method implemented in GCTA. This will help ensure that these estimates are not somehow erroneous either due to some user error or bug in the LD score software.

Response: In the previous version, we reported the heritability in observed scale for binary trait, which is not comparable across different studies. To make the information useful for researchers, we re-calculated the SNP heritability in liability scale using both LD score regression and GCTA-GREML. The results have been briefly summarized in the '**Result**' (page 4) and have also been discussed more extensively in "**Supplementary Note 3: Heritability estimation of infectious diseases**" for interested readers. The SNP heritability in liability scale was estimated to be 6% on average for most infections. The hepatitis_B now has a SNP liability scale heritability of 7.61% from LD score regression and 13.22% from GCTA-GREML. As discussed in the supplementary note, the SNP heritability is much smaller than an h^2 estimate from a twin study that includes the contributions of all genetic variation, not just common SNPs's additive contribution. Insignificant

estimation of heritability (e.g. bacterial meningitis has only 842 cases) may also represent the lack of power.

In addition, we calculated the heritability explained by the GWAS significant SNPs (index SNPs with $P < 5 \times 10^{-8}$) as previously described for case-control phenotype. For quantitative trait, we used the coefficient of determination R^2 from linear regression to calculate the phenotype variance explained by the GWAS significant associations. The heritability explained by the GWAS significant SNPs are all much less than 100% of estimated SNP heritability (**Supplementary Table 7**).

3) Greater detail should be given describing how cases and controls were defined for each study in the Methods. Why do the total number of individuals for most of the studies add up to much less than 200,000? How does exposure to vaccine (for chicken pox, measles, mumps, rubella, etc...) effect inclusion and classification in the study? ...I now see that the survey and logic for selecting cases and controls is in the Supplemental information. While this is good, as case/control definitions are central to what's actually being tested here, the overall logic should be described in the text and the supplemental referenced.

Response: Thanks for pointing this out. Our samples were drawn from more than 200,000 research participants who completed a standardized questionnaire about their history of infections and participants chose which question to answer in the entire survey. We have added more detail about how the phenotypes are defined (page 3, page 5, page 8 and page 28) and whether vaccination status are considered for the phenotypes in the main text and also in the method section, with both referencing to '**Supplementary Notes: surveys and phenotype scoring Logic**'.

4) No odds ratios are reported in the text, so effect size of these SNPs are unable to be gauged. I realize they are listed in table 2, but it is helpful not to have to look up the ones that are discussed in evaluating the possible significance of the results. Therefore, ORs should be provided in the same instances where p-values are given in the text. The authors might also want to indicate that the ORs they have found in many instances are quite large

compared to most GWAS findings to further emphasize the significance of their findings.

Response: Thanks for pointing this out. We have added OR to the text.

5) No information is given on how the actual GWAS data will be disseminated. The results of these GWAS should be made publicly available upon publication. This should at least include all rsIDs, risk alleles, p-values, and ORs for each of the 23 studies either with the online manuscript or a publicly accessible hosted webpage. In this way, identified SNPs that did not reach genome-wide significance can be further examined and other investigators can perform additional meta-analyses on this extraordinarily rich dataset. I imagine these studies could become a frequently cited resource if the data are made available in this manner.

Response: We will make the top 8k (by p value) test statistics of each GWAS and all HLA allele/amino acid test with P-value<0.05 public available in **Supplementary Data 1 and Supplementary Data 2** (page 45 Data Availability). Also researchers can request to access the full set of GWAS statistics by applying to our collaboration program. Similar efforts have been done in multiple meta-analysis collaborations/publications.

6) QQ plots should be provided to demonstrate lack of overall inflation of test statistics

Response: We have added the QQ plots of each GWAS in the **Supplemental Figure 1** and also discussed briefly in the text that there were no inflation of test observed (page 4).

Suggestions:

1) It would be informative to more comprehensively catalog how well previous GWAS of the same infectious diseases did or did not show replication. This is an opportunity to demonstrate how well replication occurs for “genome-wide significant” SNPs across several different IDs. This analysis could inform how well replication can be expected to occur and how much of a problem heterogeneity between study populations may be.

Response: Thanks for pointing this out! We have added a review of previously published GWASes and discussed the replication and the reason that we failed to

replicate on some phenotypes in **Discussion** (page 24, also summarized in **Supplementary Table 2**).

2) The GWA results for tonsillectomy were particularly interesting and the overlap with autoimmune diseases thought-provoking. If the authors wished, they could quantify the significance of the overlap and level of enrichment with autoimmune diseases but this is up to the discretion of the authors.

Response: Due the length of the paper, we did not discuss extensively the overlapping with autoimmune diseases in the main paper, but we have pointed this out in the ‘tonsillectomy’ result section (page 19-20) and discussed briefly at the end of ‘discussion’ section (page 27). We had a collaboration paper which used the similar data trying to look for genetic overlap and pleiotropic effect with a range of phenotypes including tonsillectomy and autoimmune diseases (<http://www.nature.com/ng/journal/v48/n7/abs/ng.3570.html>).

3) For variants that come up repeatedly in ID, such as *FUT2*, a diagram that depicts the directionality of this variant for risk of various diseases would be helpful and possible mechanism for this directionality could be discussed.

Response: We have created the forest plot of *FUT2* (and also for *TBX1*) in ‘**Supplementary Figure 5**’ and discussed briefly the directionality in the ‘mumps’ result section.

Minor comments:

1) Typos of p20—extra period and mistakenly italicized text

Response: Thanks!

2) P23—the authors state that “bacteria grow extracellularly” but this is not strictly true. This is certainly true of streptococcus and many other bacteria, but TB for example can grow intracellularly within macrophages, while other bacteria such as Chlamydia are obligate intracellular pathogens.

Response: We have changed the wording and reflected the possibilities that bacteria can also grow intracellularly in the discussion (page 26).

Overall, I enjoyed reading this manuscript and feel the authors have put together a nice study that will be of high interest to the ID genetics community. However, the issues listed above do substantially detract from what could be an even better manuscript and need to be corrected.

Reviewer #2 (Remarks to the Author):

This is a very large GWAS on self-reported infectious disease phenotypes for more than 200,000 European research participants. The analysis is a classic GWAS complemented by a more detailed investigation of the HLA regions because of the expected concentration of hits there.

The strength of the study is in the numbers. The weakness of the study is in the interpretation of the data (some examples below). Suboptimal interpretation occurs both in the area of infectious diseases, and also in the area of immunogenetics. It would be highly recommended that the authors would join to the study respected researchers and clinicians in those areas. This would dramatically increase the value of the report.

Response: We have invited Dr. Suzanne Noble, who is a professor working in UCSF and studies infectious diseases, to help on revising the manuscript. She re-interpreted some of the result and made many other suggestions in the discussion.

Specific comments and examples:

1. "There was a phenotypic correlation (Pearson $r = 0.41$, $P < 2.2 \times 10^{-16}$) among women who reported both UTI and yeast infections in our cohort. These two infections can have similar symptoms and both cause discomfort in vaginal area, but different infectious agents cause them." Comment: in reality, the correlation is established by the triggering of the candidae infection by the

antibiotics used to treat the bacterial infection. There is less a shared pathogenesis and genetics, than a secondary effect of treatment.

Response: Thanks for pointing it out. We have updated the discussion on the genetic correlation between UTI and yeast infection to reflect the antibiotics treatment effect. Due to the length of the paper, we moved the detail discussion of heritability and correlation analyses into Supplementary Note 2-3.

2. Tonsillectomy. The complex pattern of association with innate and acquired immunity loci is interpreted as reflecting diversity in pathogens. Heterogeneity of pathogens and mechanisms would dilute the signal – contrary to what is described. More plausibly, the results reflect the uniqueness of inflammatory response (ie., exaggerated) that may accompany the response to infectious insults.

Response: Thanks for pointing this out and we agree that heterogeneity of pathogens would actually dilute the signal. We have updated the discussion that “The large number of associations may reflect, in part, the large sample sizes for these two traits ($n = 173,412$ and $121,810$, respectively). There were many associations between both tonsillitis and middle ear infections with HLA, as well as genes with roles in the innate immune response. Lymphoid hyperreaction throughout the mucosa-associated lymphoid tissue system has been suggested as a molecular mechanism underlying the genetic association with tonsillectomy. In addition, we discovered associations between both syndromes and genes with roles in embryonic development, underscoring the important role of host anatomy in both syndromes.” (page 25)

3. Hepatitis A. The study maps (or not, according to the statistical threshold) this infection to IL28B. This is the locus of Hepatitis C. The most parsimonious explanation is that the responders are attributing their “hepatitis” experience to hepatitis A (a picornavirus) instead as to Hepatitis C (a flavivirus).

Response: Our cohorts are self-reported phenotypes and will have recall bias. It is likely that a few percent of participants reported their diagnosis wrong, which will

lower GWAS power, especially when the sample sizes are small. But it would require a significant number of participants who mixed up the answers to create a false GWAS association. As regarding to the parsimonious explanation, the association with *IL28B* seems not likely due to confused 'hepatitis C' experiences. We ran a GWAS with hepatitis C phenotype from our database and did not found an *IL28B* association with Hepatitis C due to small sample size (~2000 cases). If the hepatitis A *IL28B* association was due to a subgroup of our hepatitis A percipients who actually have hepatitis B, we would see a much stronger association in our hepatitis C GWAS. We feel that the association of Hepatitis A with *IL28B* is interesting and worth further investigation.

4. The manuscript is quite successful at trying to tease out the signals in the HLA regions. Mapping, conditional analyses, imputation of groove amino acids are all well founded steps. And somehow, at the end the reader feels that the data is not well represented, well presented graphically (showing the sharing and proximity of signals across the regions for all pathogens). The lack of solution of signals in the HCP5 regions is also a surprise, because their conditioning analysis should have supported the tagging of HLA alleles – but it is described as the contrary: the HLA regions loses statistical power when conditioned by the HCP5 alleles.

Response: We have summarized the associations and showed the proximity of HLA signals in Figure 1 (page 45). We have SNPs tagging the HCP5 region. For example, a SNP upstream of *HCP5* is significantly associated with the history of cold sores in our GWAS (page 7). There are many functional variants in the HLA region in addition to well-studied HLA classical alleles. It is very likely that variant in *HCP5* or other non-HLA classical gene close to *HCP5* playing roles in susceptibility to cold sores.

5. GWAS standards. Would not report alleles that are labelled as “moderately associated” (eg. DQB1*06:02 was moderately associated with hepatitis B ($P = 5.47 \times 10^{-6}$)). Similarly, there is some contradiction on the reports on Hepatitis A:

Table 1 lists it as “no-GWAS” hit, but it is discussed in the text, and presented as positive association in Table 2

Response: Right. We do not want to overlook interesting findings. The DQB1*06:02 association has a moderate P-value, but it has been reported in some other studies. Other researchers may use our result as a replication study that normally requires much less significance to be qualified as an interesting finding. Similarly for hepatitis A, the *IFNL4* region has been reported previously and it may be interesting to researchers in this specific diseases domain.

6. GWAS standards. While the study is announced as “European” population, and correctly using PC1-PC5 axes for population stratification control, there is a description of the study including 3% non-EUR. It may be desirable to drop the minor population of non-EUR to avoid some uncorrected structure and different epidemiology.

Response: We estimated the local ancestry for each participant. The local ancestry assignments are aggregated to give genome-wide ancestry proportion (global ancestry) for each individual in each reference population. Each participant is required to have at least 97% European ancestry (the rest 3% ancestry can belong to other ethnic group,) to form a homogeneous European cohort. For example, A Mexican American have around 50% European ancestry, he/she is not qualified to be included in European GWASs. The detail method of ancestry decomposition is described in: Durand, E. Y., Do, C. B., Mountain, J. L. & Macpherson, J. M. *Ancestry Composition: A Novel, Efficient Pipeline for Ancestry Deconvolution*. (2014).

7. GWAS standards. Figure 1. Summary of Independent HLA Signals – Unclear multiple testing correction. This is also true at other sections in the manuscript. There should be a clear statement on the study power threshold and its strict application.

Response: The conditional analysis also used stringent ‘5e-8’ cutoff. We stated it in the method: “performed forward iterative conditional regression to identify other

independent signals if the conditional p-value was $< 5 \times 10^{-8}$ " (page 32).

8. Order of the text: The initial results describe the pairwise LD score correlation on GWAS statistics. This should come at the end – once that the result of the study have been presented. Otherwise it is confusing.

Response: Yes, agree. We now briefly summarize the heritability and correlation analyses in the result summary section (page 4) and move the detail discussion into **Supplementary Note 2-3** due to the length of the paper.

9. Discussion. There is extensive repetition of results. The text should be used to highlight the high level understanding of the broad mapping across the HLA regions, and the obvious associations across the innate immunity. In addition, there should be a better discussion of previous GWAS studies. There is minimal effort at summarizing and contrasting the existing literature on infectious diseases GWAS. For example, meningococcal meningitis – common cause of meningitis, is associated with complement deficiency. Here the broader “bacterial meningitis” is associated with CA10 (carbonic anhydrase X). It may be correct – but deserves some reflection on the discrepancy – and a report on the p values for the alleles that have been previously reported.

Response: Thanks for pointing this out. We re-wrote the **discussion**: added a review of previously published GWASes (page 23-24) and also discussed the replication and the reason that we failed to replicate for some phenotypes. The discrepancy of meningococcal disease findings was also discussed. And we also highlighted the innate immunity findings and the broad mappings of HLA signals in the **discussion** (page 25).

Minor comments:

1. Fig S3 - The purple dotted line corresponds to $P=5 \times 10^{-8}$, - It does not appear to match the plot
2. “A-Val95’ means amino acid Val at position 95 of the HLA-A protein. We use this naming convention throughout the paper, which is composed from the name of the HLA protein (A), followed by the three-letter code for the amino acid (Val) and its location in the protein”. This non-conventional nomenclature is ill-advised. It may be interpreted as Ala95Val.

3. Incomplete references:

36. Learn the Signs and Symptoms of TB Disease.

45. Pneumococcal Disease.

48. Medical microbiology. (University of Texas Medical Branch at Galveston, 1996).

78. Ear Infections in Children.

89. Sore Throat.

Response: We have updated the HLA protein nomenclature to use '*HLA-A Val95*' representing amino acid Val at position 95 of *HLA-A* protein and we have updated the references.

REVIEWERS' COMMENTS:

Reviewer #1 (Remarks to the Author):

This resubmitted manuscript on GWAS of 23 infectious diseases is substantially improved. My previous concerns have been well addressed:

- 1) The LD score regression results are presented more accurately now with the negative (though non-significant) correlations no longer set to 0 and are likely more accurate with removal of poorly imputed SNPs.
- 2) Heritability analyses have been greatly expanded with inclusion of GCTA estimation of SNP-based heritabilities and estimation of SNP contribution to h^2 . I found these new sections interesting to read--the low heritability is quite different from autoimmune diseases and other common polygenic human diseases and still the GWAS hits are often quite robust.
- 3) Greater care has been taken in describing definitions of cases and controls.
- 4) Greater access to data is now provided.
- 5) The comparison to previously published GWAS to test how well replication of hits occurs has been added. Making these comparisons is valuable to the field and nicely presented. Interesting how even many highly significant GWAS results did not replicate, and I appreciated the brief discussion of factors and limitations of GWAS and self-reporting that may play into this.
- 6) Forest plot for FUT2 is a nice addition.

I also thought that Reviewer 2's comments were spot on and feel that the changes/additions made to address his/her concerns were also well done and improve the quality of the manuscript.

A few remaining minor errors:

- 1) MAGENTA is misspelled in Table 4, 5, 6.
- 2) There appear to be multiple errors in referencing Figures, Tables, Supplemental materials in the text. Please recheck numeration.

Dennis Ko, MD PhD
Duke University

Reviewer #2 (Remarks to the Author):

The authors have given satisfactory responses to the queries

Reviewer #1 (Remarks to the Author):

A few remaining minor errors:

1) MAGENTA is misspelled in Table 4, 5, 6.

Response: Thanks for pointing this out. We have corrected the mistakes.

2) There appear to be multiple errors in referencing Figures, Tables, Supplemental materials in the text. Please recheck numeration.

Response: Thanks for pointing this out. We have identified and corrected the numeration.